# Discovery of a selective alpha-kinase 1 inhibitor for the rare genetic disease ROSAH syndrome

Jieqing Fan[1,2,7], Danyang Liu[1,2,7], Zhu Ming[1], Chunyu Yan[1], Huaixin Dang[2], Yanfang Pan[1], Xiong Wei[1], Zhengle Zhao[1], Wenzhi Wang[1], Shuai Zhang[2], Linlin Chen[1], Shuo Cai[1], Jiangbin Ke[3], Yaru Luo[3], Linjie Rao[3], Jingjing Chen[2], Zhenjie Chen[1], Junlin Zhou[1], Feixiang Chen[1], Xiaodi Duan[2], Boyue Ren[1], Tong-Ruei R. Li[1], Lawrence Melvin[4], Jeysen Yogaratnam[4], Vinit B. Mahajan[5], Hongmei Song[6], Henri Lichenstein[4], Tian Xu[1] ✉ & Cong Xu [1,2] ✉

ROSAH (retinal dystrophy, optic nerve edema, splenomegaly, anhidrosis, and headache) syndrome is a rare genetic disease caused by variants in alpha-kinase 1 (ALPK1) resulting in downstream pro-inflammatory signaling mediated by the TIFA/TRAF6/NF-κB pathway. Here, we report the design of an ALPK1 inhibitor, DF-003, with pharmacokinetic properties suitable for daily oral dosing. In biochemical assays, DF-003 potently inhibits human ALPK1 ($IC_{50} = 1.5$ nM) and the ROSAH disease-causing mutant ALPK1[T237M] ($IC_{50} = 16$ nM). When tested against a panel of 394 human kinases, DF-003 exhibits ≥860-fold selectivity over the closest kinase. In cell-based assays, DF-003 suppresses inflammatory cytokine signaling mediated both by wild-type ALPK1 and the disease-causing ALPK1[T237M] mutant. Using mice heterozygous for wild-type human *ALPK1* and *ALPK1[T237M]* established to model ROSAH syndrome that exhibit retinal microglial infiltration, astrocyte activation, and inflammatory cytokine upregulation in the retina, optic nerve, and cortex, we show that orally administered DF-003 is sufficient to inhibit these inflammatory phenotypes.

ROSAH (retinal dystrophy, optic nerve edema, splenomegaly, anhidrosis, and headache) syndrome is a rare, autosomal dominant autoinflammatory disease named according to the characteristic symptoms exhibited by affected patients[1,2]. The most common presenting symptom is a progressive decline in visual acuity that typically begins before 20 years of age, with ophthalmologic examination often revealing optic disc elevation, uveitis, and retinal nerve degeneration[2,3]. ROSAH patients also exhibit inflammatory features such as non-infectious low-grade fevers, arthralgia, headaches, and persistently elevated levels of serum inflammatory cytokines including tumor necrosis factor α (TNF-α), interleukin-6 (IL-6), and IL-1β[3]. ROSAH has been shown to be caused by at least two heterozygous missense variants in the gene encoding alpha-kinase 1 (*ALPK1*), p.Thr237Met (T237M) and p.Tyr254Cys (Y254C), with the T237M disease-causing mutation being the most frequently reported[2,3].

[1]Shanghai Yao Yuan Biotechnology Ltd (Drug Farm), Shanghai, China. [2]Zhejiang Yao Yuan Biotechnology Ltd, Jiashan, Zhejiang, China. [3]Oujiang Laboratory (Zhejiang Lab for Regenerative Medicine, Vision and Brain Health), Wenzhou, Zhejiang, China. [4]Drug Farm USA LLC, Guilford, CT, USA. [5]Molecular Surgery Laboratory, Byers Eye Institute, Stanford University, Palo Alto, CA, USA. [6]Department of Pediatrics, Peking Union Medical College Hospital, Chinese Academy of Medical Sciences and Peking Union Medical College, Beijing 100730, China. [7]These authors contributed equally: Jieqing Fan, Danyang Liu. ✉e-mail: tian.xu@drugfarminc.com; tony.xu@drugfarminc.com

ALPK1 is a member of the atypical human alpha-kinase family and functions as an innate immune pattern recognition receptor (PRR) for ADP-heptose, a soluble byproduct of the metabolic processing of bacterial lipopolysaccharide (LPS)[4–6]. ADP-heptose sensing by ALPK1 triggers its phosphorylation and the direct ALPK1-mediated phosphorylation of multiple threonine residues in the adapter protein TIFA[4,7]. TIFA, in turn, oligomerizes to form TIFAsomes that trigger TRAF6 multimerization and the activation of the proinflammatory transcription factor NF-κB[4,5,7]. Recently, ALPK1 was also shown to be activated by similar pathogen-associated molecular patterns (PAMPs) across multiple kingdoms, including cytidine diphosphate (CDP)-heptose and uridine diphosphate (UDP)-heptose, both capable of initiating downstream NF-κB-mediated inflammatory signaling[8]. Consistent with its role as a PRR, ALPK1-dependent innate immunity has been observed for several bacterial infections[4,9–11], and the small molecule-mediated agonism of ALPK1 signaling can protect against hepatitis B virus (HBV)[12]. Both the p.T237M and p.Y254C ALPK1 mutations increase basal NF-κB activity in reporter cells[3], and the former also enhances ALPK1 autophosphorylation and TIFA phosphorylation[7]. Recently, Snelling et al. reported that the p.T237M

variant expands the range of ligands to which the mutant enzyme is responsive such that it can be activated by endogenous nucleotide sugars including UDP-mannose that fail to activate wild-type ALPK1[13]. Disease-causing mutations in *ALPK1* give rise to the inflammatory phenotypes exhibited by individuals with ROSAH syndrome[3]. A recent paper has also linked the ADP-heptose-ALPK1 axis to ageing and expansion of rare leukemic cells, thus suggesting ALPK1 as a potential target to prevent progression of clonal haematopoiesis of indeterminate potential (CHIP) to overt leukemia and other related immune conditions[14].

At present, there is no approved drug for the treatment of ROSAH patients. Drugs that antagonize cytokines (IL-6, TNF-α, IL-1β) including tocilizumab, adalimumab, anakinra, and canakinumab have all been administered off-label to ROSAH patients with reports of varying levels of efficacy[3,15–17]. Specifically, early clinical data has shown that tocilizumab may have a beneficial effect in reducing ocular inflammation in ROSAH patients[3].

In this work, given the established link between ALPK1 disease-causing mutations and autoinflammatory ROSAH phenotypes, we have developed an ALPK1 inhibitor, DF-003 to treat deleterious outcomes associated with dysregulated ALPK1 activity. DF-003 exhibits good potency and selectivity for ALPK1 over other human kinases, specifically blocking ALPK1- and p.T237M-induced cytokine upregulation in vitro. In a mouse model of ROSAH syndrome generated through the knock-in of heterozygous human *ALPK1* harboring the p.T237M mutation, DF-003 inhibits upregulation of inflammatory cytokines in the retina, optic nerve, and cerebral cortex. DF-003 also suppresses increases in microglial infiltration and astrocyte activation in the retina. These results offer a strong rationale for human clinical trials testing DF-003 as an ALPK1 inhibitor with the potential to serve as a precision-targeted drug for treating the progressive manifestations of ROSAH syndrome.

## Results

### DF-003 is a potent and selective inhibitor of ALPK1

The discovery of DF-003 (Fig. 1a) was the result of convergent structure-activity relationship (SAR)-focused design. For full details regarding the design and synthesis of DF-003, and the molecular modeling-based characterization of DF-003, see Supplementary Notes 1–2. We evaluated the activity of DF-003 in a radiolabeled kinase assay in which DF-003 dose-dependently inhibited ALPK1 kinase activity with an $IC_{50}$ of 1.5 nM (Fig. 1b). DF-003 also exhibited good selectivity for ALPK1 when tested against a panel of 394 human kinases (Supplementary Data 2). The $IC_{50}$ values for the three non-ALPK1 kinases against which DF-003 exhibited the highest levels of activity were 1.29 μM, 3.10 μM, and 4.60 μM for TAOK2/TAO1, CAMK2g, and CAMK1a, respectively (Fig. 1c). To evaluate whether DF-003 is an ATP-competitive kinase inhibitor, we conducted a biochemical assay using ALPK1 with varying concentrations of ATP. The results demonstrated that the $IC_{50}$ of DF-003 increased as the ATP concentration increased, indicating that DF-003 functions as an ATP-competitive kinase inhibitor (Supplementary Fig. 1).

### DF-003 inhibits ALPK1-mediated cytokine/chemokine production

ADP-heptose (showing D-isomer in Fig. 2a) is well-established as a canonical PAMP that triggers ALPK1 activation[4–6], serving as an essential mediator for the upregulation of inflammatory cytokines and chemokines including TNF-α and CXCL8 (also known as IL-8) in response to bacterial infection[11]. We previously developed the ADP-heptose derivative DF-006 (Fig. 2a) as a serum-stable small molecule ALPK1 agonist that triggers TIFA phosphorylation and the upregulation of *Tnf* and other proinflammatory genes[12]. Once phosphorylated, TIFA dimers engage in intermolecular interactions and trigger the assembly of large TIFAsomes[9]. DF-003 potently suppressed DF-006-

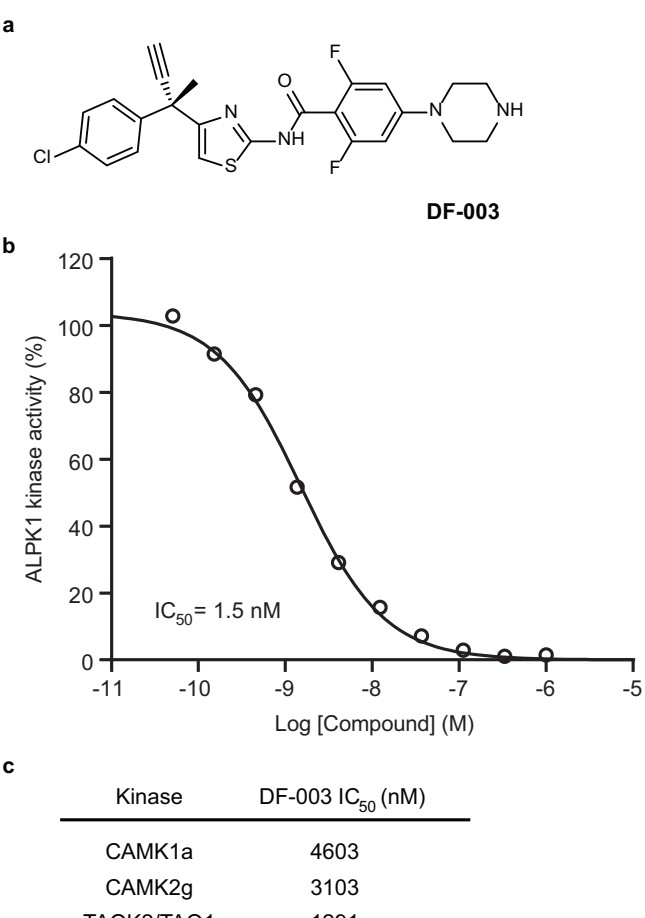

**a**

**DF-003**

**b**

IC$_{50}$= 1.5 nM

(Graph: y-axis "ALPK1 kinase activity (%)" from 0 to 120; x-axis "Log [Compound] (M)" from -11 to -5)

**c**

| Kinase | DF-003 IC$_{50}$ (nM) |
|---|---|
| CAMK1a | 4603 |
| CAMK2g | 3103 |
| TAOK2/TAO1 | 1291 |

**Fig. 1 | DF-003 is a potent and selective inhibitor of ALPK1. a** Chemical structure of DF-003. **b** An in vitro kinase assay was performed using recombinant human ALPK1, human TIFA, and ADP-D-heptose in a kinase reaction buffer containing 3-fold dilutions of DF-003. After adding [$^{33}$P]-ATP, the kinase reaction was allowed to proceed at room temperature for 60 min, after which ALPK1 kinase activity was quantified and expressed as the percentage of remaining ALPK1 kinase activity relative to vehicle (DMSO, no DF-003) conditions. Each condition was measured without replication but the assay was conducted 4 times with similar results. **c** The top 3 non-ALPK1 human kinases against which DF-003 was found to exhibit the most potent inhibitory activity in an in vitro kinase assay-based screen of 394 human kinases are presented with calculated DF-003 IC$_{50}$ values.

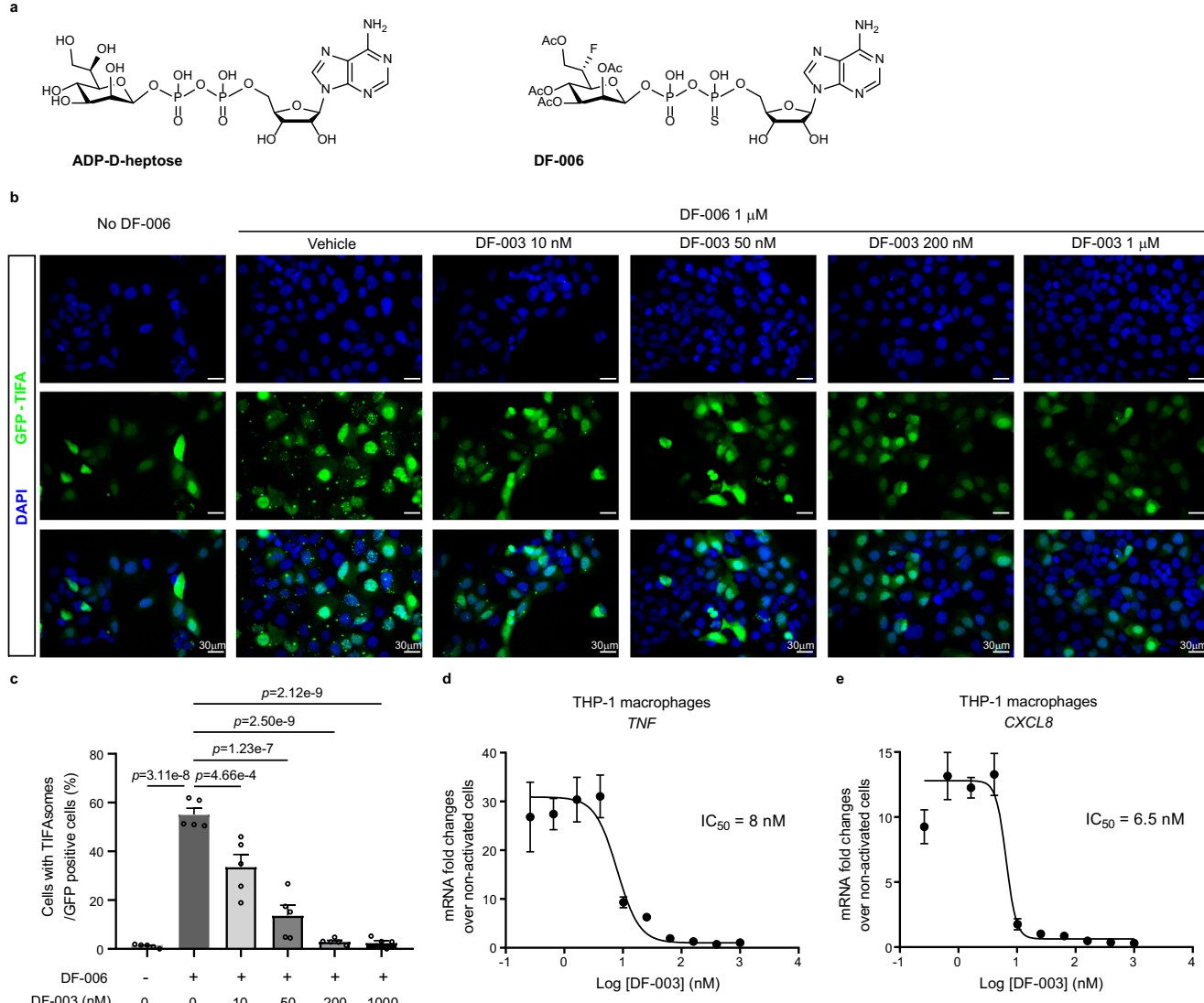

**Fig. 2 | DF-003 blocks induction of TIFAsome formation and chemokine/cytokine production by the ALPK1 agonist DF-006. a** Chemical structures of the ALPK1 agonists ADP-heptose (showing the D-isomer, ADP-D-heptose) and DF-006. **b** HEK-293 cells were transfected with GFP-TIFA overexpression plasmids. Cells were treated with 10 nM to 1 μM DF-003 or vehicle followed by 1 μM DF-006 stimulation for 1 h. Nuclei are labeled with DAPI (blue), while TIFA is labeled with GFP, and TIFAsomes are represented by punctate green fluorescent structures. **c** Quantification of the fractions of cells with TIFAsomes among total GFP positive cells as shown in (**b**). **d, e** THP-1 cells were treated with PMA for 48 h to induce macrophage differentiation, after which they were incubated for 2 h with serial dilutions of DF-003 dissolved in DMSO. DF-006 was then added for 4 h, and qPCR was used to measure *TNF* (**d**) and *CXCL8* (**e**) mRNA expression, with *GAPDH* serving as a normalization control. Data were expressed as mRNA fold induction compared to THP-1 macrophages not treated with DF-006 activation. Curve fitting and IC$_{50}$ value calculations were performed with GraphPad Prism 6. In (**c**) data represent means ± SEM of 5 technical repeats from 2 independent studies. Statistical comparisons between non-DF-003 treated cells with or without DF-006 were made using two-tailed unpaired Student's *t*-tests and comparisons between DF-003-treated groups to the vehicle-treated, DF-006-stimulated group were made with one-way ANOVAs followed by Dunnett's post hoc tests. In **d, e** data represent means ± SEM for the quadruplicate technical repeat wells at each treatment dose.

induced TIFAsome formation (Fig. 2b, c), suggesting its intracellular inhibition of ALPK1. We also tested the ability of DF-003 to suppress the induction of inflammatory cytokine and chemokine gene expression in THP-1 macrophages triggered by DF-006. DF-003 exhibited IC$_{50}$ values of 8 nM and 6.5 nM for the inhibition of DF-006-induced *TNF* and *CXCL8* upregulation, respectively (Fig. 2d, e). These data demonstrate that DF-003 disrupts inflammatory signaling downstream of ligand-mediated ALPK1 activation in immune cells.

## DF-003 inhibits ALPK1[T237M]-mediated cytokine/chemokine production

To assess whether DF-003 was also able to inhibit the disease-causing ALPK1[T237M] mutated enzyme, we performed an in vitro kinase assay with recombinant human TIFA and ALPK1[T237M] in the presence or absence of ADP-D-heptose or UDP-mannose, a reported endogenous ligand for ALPK1[T237M][13]. We observed that ALPK1[T237M] was inactive in the absence of these exogenous activating ligands, whereas it phosphorylated TIFA in a time-dependent manner in response to both ADP-D-heptose and UDP-mannose (Fig. 3a). DF-003 inhibited UDP-mannose-stimulated ALPK1[T237M] in a dose-dependent manner with an IC$_{50}$ of 16 nM (Fig. 3b).

We next assessed whether DF-003 could inhibit ALPK1[T237M] and associated downstream signaling pathways in human cells. To that end, we established NF-κB reporter cells overexpressing (OE) ALPK1 or ALPK1[T237M]. We observed enhanced NF-κB reporter activity in the absence of exogenous stimulation in the

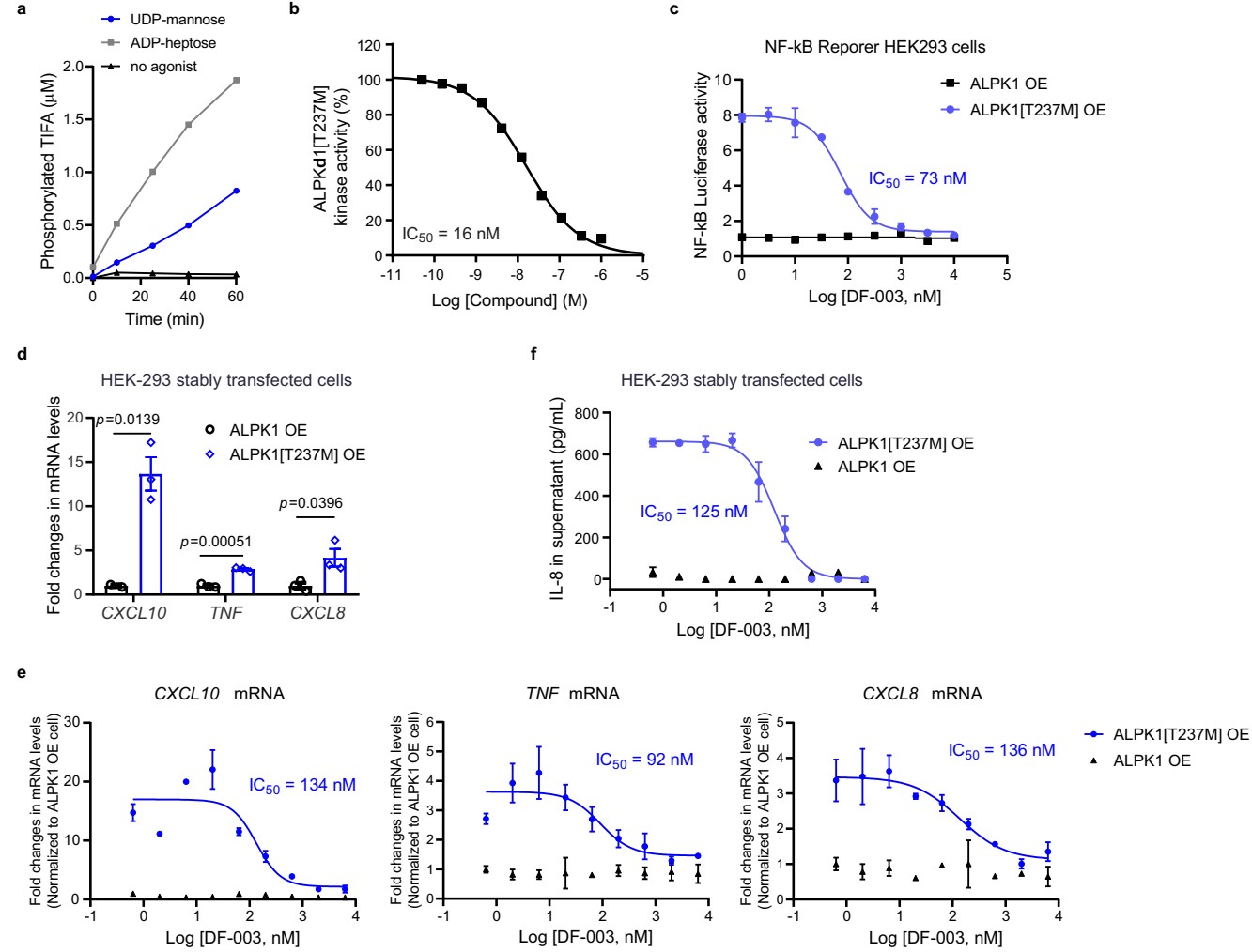

**Fig. 3 | DF-003 inhibits the kinase activity of ALPK1[T237M] and blocks NF-κB target gene expression induced by ALPK1[T237M]. a, b** A [$^{33}$P]-ATP in vitro kinase assay performed without inhibitor (**a**) or with DF-003 (**b**). In (**a**) each dot represents a single well measurement. In (**b**) ALPK1[T237M] was agonized by UDP-mannose and the kinase activity was expressed as the percentage of remaining ALPK1[T237M] kinase activity relative to vehicle (DMSO, no DF-003) conditions. Each dot represents the mean of duplicate technical repeats. **c** NF-κB reporter cells were transiently transfected with plasmids to overexpress human ALPK1 or ALPK1[T237M]. Four hours after transfection, cells were treated with DF-003 for 48 h, and intracellular firefly luciferase activity was measured to evaluate NF-κB activation. **d, e** HEK-293 cell lines stably overexpressing (OE) ALPK1 or ALPK1[T237M] were treated with serially diluted DF-003 for 30 h, after which *CXCL10*, *TNF*, and *CXCL8* mRNA levels were measured by qPCR and normalized to levels of *GAPDH*. **d** Fold induction of *CXCL10*, *TNF*, and *CXCL8* mRNA levels in vehicle (DMSO)-treated ALPK1[T237M] OE cells compared to those in ALPK1 OE cells. **e** Inhibition of *CXCL10*, *TNF*, and *CXCL8* mRNA upregulation in ALPK1 OE and ALPK1[T237M] OE cells treated with DF-003. **f** ALPK1 OE and ALPK1[T237M] OE HEK-293 cells were treated with serially diluted DF-003 for 30 h, with the replenishment of media and DF-003 at 24 h. Supernatant CXCL8 levels were then measured by ELISA. In **c**–**f** data represent the mean ± SEM for the triplicate wells of each treatment group. Statistical outliers were determined using Grubb's test and indicated in the Source Data file. In (**d**) statistical comparisons to the wild-type ALPK1-overexpressing stable cell line group were made using the two-tailed unpaired Student's *t*-tests. In (**b**–**f**) curve fitting and IC$_{50}$ value calculations were performed with GraphPad Prism 6.

ALPK1[T237M] OE cells, and DF-003 treatment suppressed this activity in a dose-dependent manner (Fig. 3c). In HEK-293 cells, overexpression of ALPK1[T237M] increased *TNF*, *CXCL10*, and *CXCL8* expression relative to the levels of these mRNAs in cells overexpressing wild-type ALPK1 (Fig. 3d). DF-003 dose-dependently inhibited cytokine upregulation in HEK-293 cells overexpressing ALPK1[T237M] and had no impact on the basal expression of *TNF*, *CXCL10*, or *CXCL8* in cells overexpressing wild-type *ALPK1* (Fig. 3e). Consistent with these changes at the mRNA level, DF-003 suppressed (IC$_{50}$ = 125 nM) the enhanced secretion of CXCL8 from HEK-293 cells overexpressing ALPK1[T237M] (Fig. 3f). DF-003 is thus able to potently inhibit ALPK1[T237M]-induced TIFA phosphorylation and downstream inflammatory cytokine and chemokine production in human cell lines.

## Evaluation of the pharmacokinetic properties of DF-003

To support the in vivo use of DF-003 for ROSAH syndrome, we analyzed its pharmacokinetic properties in wild-type C57BL/6J mice (Supplementary Fig. 2, Supplementary Table 2). Oral administration of DF-003 at 3 or 5 mg/kg once per day for 10 days was associated with a dose-dependent increase in plasma drug concentrations at all time points from 0-24 h post the last (10$^{th}$) dosing (Supplementary Fig. 2a). Maximum plasma concentrations (C$_{max}$) of DF-003 rose approximately proportionally with administered dose to 160 and 310 ng/mL in the respective treatment groups. DF-003 had an average time to C$_{max}$ (T$_{max}$) ranging from 2–2.67 h following the 10$^{th}$ dose (Supplementary Table 2). The measured half-life (t$_{1/2}$) values for DF-003 in the 3 and 5 mg/kg groups following the 10$^{th}$ dose were 7.69 and 8.81 h, respectively. Orally administered DF-003 also led to a dose-dependent

increase in the retinal and optic nerve concentrations after the 10th dose (Supplementary Fig. 2b), indicative of a drug that crosses the blood-retina barrier. High concentrations of DF-003 were also detectable in the cerebral cortex of mice from both dosing groups after the 10th dose, indicative of a drug that also crosses the blood-brain barrier (Supplementary Fig. 2b).

### DF-003 inhibits retinal microglia and astrocyte activity in ROSAH mice

In an effort to establish a model of ROSAH syndrome, we utilized *Alpk1[-/-]* mice generated previously in which the kinase function of murine ALPK1 had been disrupted by exon 13 deletion in the mouse gene[12]. CRISPR and homologous recombination were used to introduce the coding sequence of human ALPK1 (hALPK1) or hALPK1[T237M] immediately following the start codon in the murine *Alpk1-* allele, with the latter containing the same C > T nucleotide substitution as in human patients bearing the T237M disease-causing mutation. This enabled the generation of homozygous *hALPK1* knock-in (hALPK1-KI) mice and mice heterozygous for the *hALPK1* and *hALPK1[T237M]* alleles (hALPK1[T237M]-KI mice) (Fig. 4a).

We evaluated changes in retinal immune cell activation in hALPK1[T237M]-KI mice and the ability of DF-003 to impact those changes. hALPK1[T237M]-KI mice exhibited a significant increase in the numbers of detected microglia (IBA1[+] cells[18]) in both the outer nuclear layer (ONL) and inner nuclear layer (INL) of the retina as compared to hALPK1-KI control mice, and DF-003 (3 mg/kg) suppressed 68% and 53% of this microglial activation in the INL and ONL, respectively (Fig. 4b, c). Interestingly, astrocyte activation was similarly detected in retinal cross-sections from these mice by immunohistochemical staining for glial fibrillary acidic protein (GFAP)[19]. A significant increase in the frequency of GFAP positivity was detected in the nerve fiber layer of ROSAH model mice as compared to *ALPK1*-KI controls, and the administration of DF-003 (3 mg/kg) suppressed 49% of the increase in astrocyte activity (Fig. 5).

### DF-003 inhibits inflammatory cytokine upregulation in ROSAH mice

Because ROSAH is known to be an autoinflammatory disease affecting the eyes and brain, we assessed whether hALPK1[T237M]-KI mice exhibited cytokine upregulation in these clinically relevant tissues. In the retina, *Ccl2, Ccl5, Cxcl1, Cxcl9, Cxcl10, Tnf, Il6*, as well as the microglia marker genes *Cx3cr1* and *Aif1*, were all upregulated in hAPLK1[T237M]-KI mice (Fig. 6a), and daily oral administration of DF-003 for 10 days suppressed the upregulation of these genes. Strikingly, *Cxcl10, Ccl2*, and *Ccl5* were upregulated by 305-, 102-, and 102-fold in hALPK1[T237M]-KI mice relative to hALPK1-KI controls, and DF-003 (5 mg/kg) suppressed their induction by 31%, 40%, and 50%, respectively. Treatment with 5 mg/kg DF-003 also inhibited the upregulation of *Il6* in the retinas of these mice by 56%. The optic nerves of ROSAH model mice exhibited 4.5- and 41-fold increases in the expression of *Ccl2* and *Cxcl10*, and DF-003 (5 mg/kg) suppressed their induction by 66% and 91%, respectively. These changes were not restricted to the retina or optic nerve, as 15-, 4.4-, 15-, and 196-fold increases in *Ccl2, Ccl5, Cxcl1*, and *Cxcl10* expression were also observed in cortical samples from the brains of hALPK1[T237M]-KI mice relative to hALPK1-KI controls (Fig. 6a), and DF-003 (5 mg/kg) suppressed their corresponding induction by 71%, 56%, 49% and 55%, respectively. These results demonstrate the ability of DF-003 to inhibit ALPK1[T237M]-induced cytokine expression in this mouse model of ROSAH syndrome. In general, DF-003 treatments that resulted in the downregulation of cytokine expression were well-tolerated, as no significant changes in body weight were observed throughout the course of these experiments (Supplementary Fig. 3).

### Phenotypic evaluation of ROSAH model mice

Given that ROSAH mice exhibited evidence of excessive eye inflammation and retinal damage that are hallmarks of ROSAH syndrome, we next tested whether vision in our ROSAH model mice was adversely affected at 4 and 9 months of age. Using an OptoDrum system[3], we detected no significant differences in visual acuity between hALPK1-KI and hALPK1[T237M] mice at either of the analyzed ages (Supplementary Fig. 4a). Similarly, fundoscopy and optical coherence tomography (OCT[3]) did not reveal any evidence of retinal degeneration or optic nerve/optic disc/retina edema in these ROSAH mice (Supplementary Fig. 4b, c). We also tested whether other phenotypes observed in ROSAH patients could be seen in ROSAH model mice and found that neither splenomegaly nor evidence of anhidrosis was evident in these mice at ages of up to 6 months (Supplementary Fig. 5a, b). Cytokines and chemokines frequently upregulated in human ROSAH patient plasma were also not elevated in plasma samples from hALPK1[T237M]-KI heterozygous mice, except for CXCL9 (Supplementary Fig. 5c).

## Discussion

At present, there is no approved drug for ROSAH syndrome, and patients have been typically treated with anti-TNF and IL-1 inhibitors[3,15–17]. These drugs have been linked to improvements in subjective symptoms of ROSAH, while tocilizumab (anti-IL-6) showed early promising data with reduced ocular inflammation in patients unresponsive to TNF and IL-1 inhibition[3]. However, formal clinical trials of drugs to treat ROSAH have not yet been initiated, and with limited data, it remains uncertain whether the positive effects of individual anti-cytokine therapy are attributable to drug efficacy or inherent disease variation. Because the role of individual cytokine inhibitors remains untested, we sought to develop a drug that targets the root cause of ROSAH and has the potential to inhibit the production of multiple cytokines caused by disease-causing ALPK1 mutations, with the ultimate goal of bringing this drug forward for testing in a clinical trial.

This report describes the identification and preclinical evaluation of a potent ALPK1 inhibitor, DF-003. In a biochemical screen of 394 human kinases, we demonstrated that DF-003 has 860-fold selectivity over the closest kinase. Thus, DF-003 is expected not to have off-target effects on other kinases at efficacious doses. DF-003 was also capable of inhibiting ALPK1 in human cells as evidenced by the ability to potently suppress wild-type ALPK1-induced inflammatory gene expression in THP-1 macrophages. Importantly for the treatment of ROSAH syndrome, DF-003 was also able to suppress ALPK1[T237M] activity in biochemical and cell-based assays.

There is conflicting data as to whether T237M disease-causing mutants result in a constitutively active ALPK1 or whether the mutation activates ALPK1 by another mechanism. In one report, ALPK1[T237M] failed to induce spontaneous TIFA phosphorylation when over-expressed in HEK-293 cells, although it was associated with a possible, albeit nonsignificant trend towards increased spontaneous TIFAsome assembly over wild-type ALPK1[7]. In contrast, an earlier report described constitutive NF-κB activation in reporter cells expressing ALPK1[T237M][3]. Snelling et al.[13] recently demonstrated the ability of endogenous nucleotide sugars including UDP-mannose to activate ALPK1[T237M] but not wild-type ALPK1, suggesting that the ROSAH-related mutation sensitizes mutant ALPK1 to activation by endogenous nucleotide sugars rather than causing its constitutive activation. Our biochemical data showed that ALPK1[T237M] was inactive unless stimulated with UDP-mannose or ADP-D-heptose, and that DF-003 could inhibit the activated recombinant mutant enzyme. Combined with our observation that inflammatory cytokines (*CXCL10, TNF, CXCL8*) were upregulated even in the absence of exogenous stimulation in HEK-293 cells, our data suggest that endogenous nucleotide sugars such as UDP-mannose or other factors within these cells

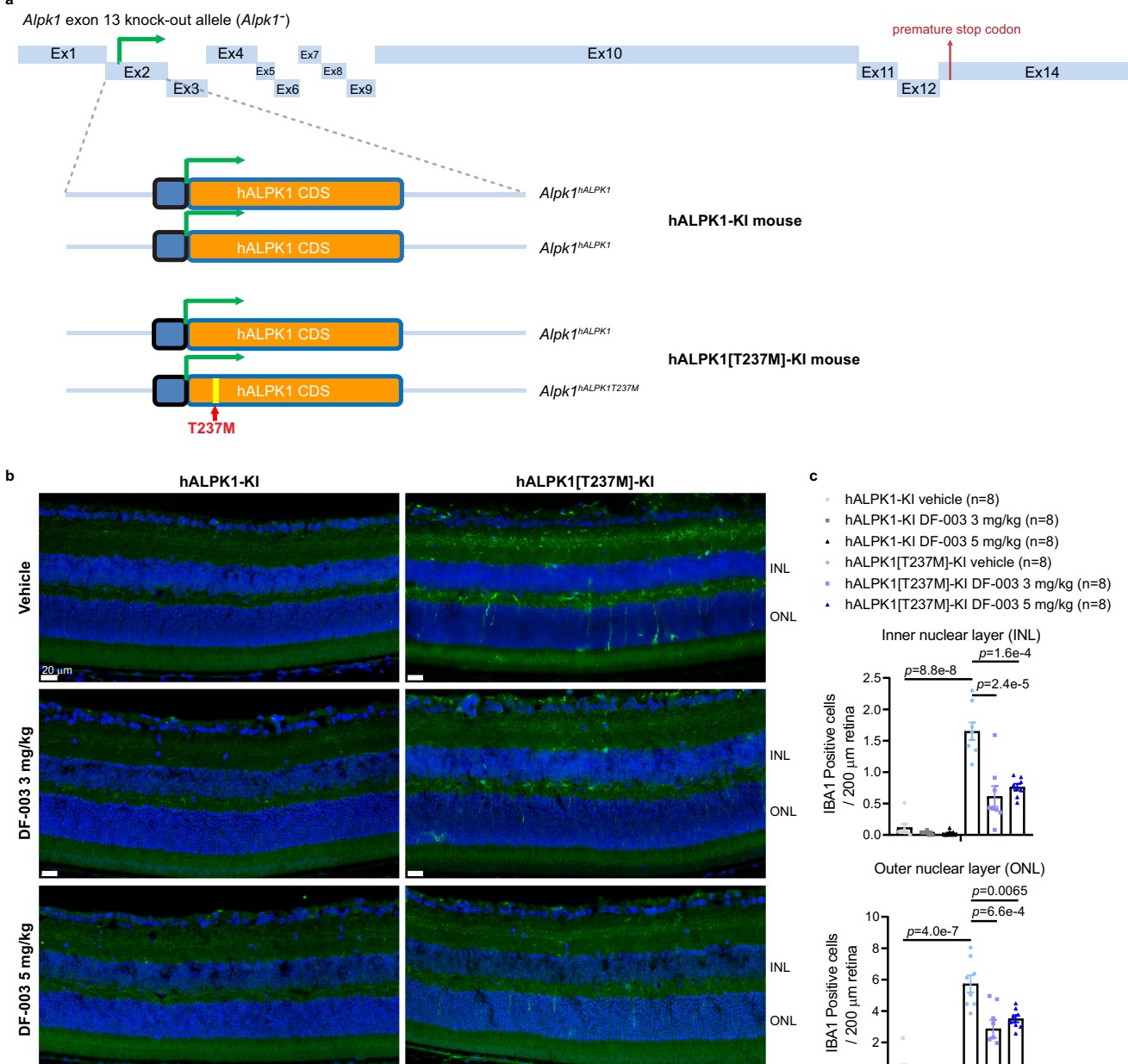

**Fig. 4 | DF-003 reduces microglial infiltration in the retinas of ROSAH model mice. a** Schematic of *Alpk1hALPK1* and *Alpk1hALPK1T237M* alleles. Ex: exon; CDS: coding sequence. Generation of the two alleles was described in Methods. Mice homozygous for *Alpk1hALPK1* were designated as hALPK1-KI mice, while mice heterozygous for *Alpk1hALPK1* and *Alpk1hALPK1T237M*, mimicking the genetic profile of ROSAH patients, were designated as hALPK1[T237M]-KI mice. **b**, **c** Female hALPK1-KI and hALPK1[T237M]-KI mice were treated orally with DF-003 once per day for 10 days at the indicated doses (*n* = 8 animals for each group). **b** At endpoint, anti-IBA1 antibody was used to label microglia on retinal cross-sections (green). **c** IBA1-positive cells in the outer nuclear layer (ONL) and inner nuclear layer (INL) of the retina from each mouse were quantified and normalized to the length of the curvature of the retina. Data represent means ± SEM of each genetic and treatment group, with each dot representing one section from one animal. Statistical comparisons between vehicle groups of the two genotypes were made using two-tailed unpaired Student's *t*-tests: in INL, F = 5.740, t = 10.05, $r^2$ = 0.88, 95% confidence interval (CI) 1.21 to 1.86, df = 14, *p* = 0.000000088; in ONL, *F* = 3.759, t = 8.879, df = 14, $r^2$ = 0.85, 95% CI 4.11 to 6.72, *p* = 0.00000040; and comparisons between DF-003 dosed groups and vehicle group within the same genotype were made with one-way ANOVAs followed by Sidak's post hoc tests: in the INL of hALPK1[T237M]-KI mouse retinas, *t* = 5.69, df = 21, 95% CI 0.59 to 1.48, *p* = 0.000024 between the vehicle and 3 mg/kg groups, and t = 4.87, df=21, 95% CI of 0.45 to 1.33, *p* = 0.00016 between the vehicle and 5 mg/kg groups; in the ONL of hALPK1[T237M]-KI mouse retinas, *t* = 4.28, df = 21, 95% CI 1.25 to 4.47, *p* = 0.00066 between the vehicle and 3 mg/kg groups, and *t* = 3.32, df = 21, 95% CI of 0.61 to 3.83, *p* = 0.0065 between the vehicle and 5 mg/kg groups.

can activate ALPK1[T237M] activity in a manner which can be antagonized by DF-003. These results offer evidence demonstrating the ability of an ALPK1 inhibitor to counteract the activity of disease-causing mutant ALPK1[T237M].

For preclinical and clinical testing, it is important that DF-003 has optimized pharmacokinetic properties, including a suitable half-life and the ability to reach retinal and brain compartments which are affected in ROSAH syndrome. In wild-type mice with the same background as the ROSAH model animals, DF-003 had a half-life appropriate for once-daily dosing and was detected in the retina and brain, indicating that it can cross the brain-retina and blood-brain barriers. To test DF-003 in vivo, we established a ROSAH mouse model that was

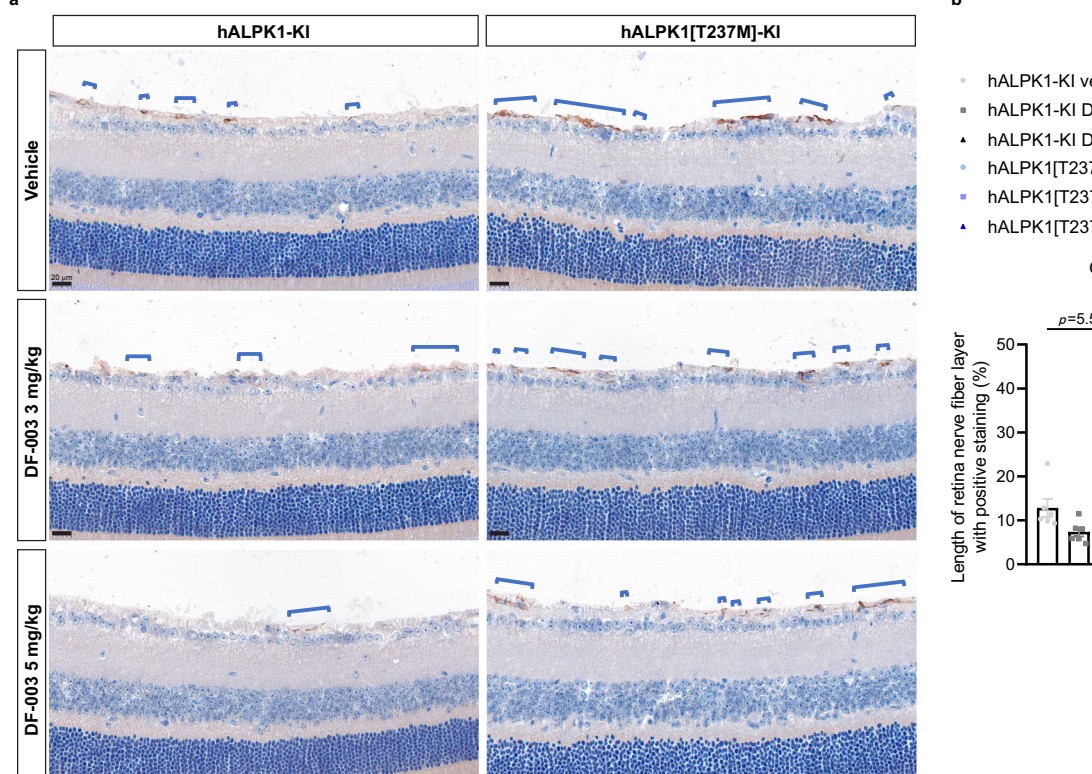

**Fig. 5 | DF-003 decreases astrocyte activation in the retinas of ROSAH model mice.** Female hALPK1-KI and hALPK1[T237M]-KI mice were treated orally with DF-003 once per day for 10 days at the indicated doses ($n = 8$ animals for each group). **a** After dosing was complete, retinal cross-sections were collected and stained with anti-GFAP antibody to label activated astrocytes via immunohistochemistry. The signal was mainly observed in the nerve fiber layer of the retina. **b** The percentage of the retinal length of the nerve fiber layer that was positive for GFAP-labeled astrocyte fibers was quantified. Data represent means ± SEM for each genetic and treatment group, with each dot representing the measurement of one section from one animal. In the first 5 groups, there were 1–2 animals excluded from the analyses due to unsuccessful processing of retina samples, which affected the integrity of GFAP staining in the nerve fiber layer. Statistical comparisons between vehicle groups of the two genotypes were made using two-tailed unpaired Student's $t$-tests: F = 1.646, $t = 8.139$, df = 11, $r^2 = 0.8576$, 95% confidence interval (CI) 19.62 to 34.16, $p = 0.0000055$; and comparisons between DF-003 dosed groups and the vehicle group within the same genotype were made with one-way ANOVAs followed by Sidak's post hoc tests: $t = 3.55$, df = 19, 95% CI 4.18 to 22.23, $p = 0.0043$ between vehicle and 3 mg/kg group; $t = 3.87$, df = 19, 95% CI 5.21 to 22.69, $p = 0.002$ between vehicle and 5 mg/kg group.

designed to mimic the human genetic disease as heterozygous carriers of one copy of human $ALPK1^{T237M}$ and one copy of wild-type human $ALPK1$. Kozycki et al. also previously generated knock-in mice bearing the p.T237M mutation in the murine $Alpk1$ gene[3], and while their mice exhibited elevated serum concentrations of chemokines including CCL2, CXCL1, and CXCL10, no changes in spleen size/weight or retinal degeneration were observed through 12 months of age. Similarly, our mice did not exhibit any apparent symptoms of splenomegaly, anhidrosis[20], or vision deficits and it remains to be determined if these deficits can occur in our ROSAH mouse model at an advanced age. Unlike the mouse model reported previously by Kozycki et al.[3], we were largely unable to detect elevated levels of these same chemokines in systemic circulation with the exception of CXCL9, the upregulation of which was evident in the serum of ROSAH model mice and suppressed by DF-003. This difference may be related to the fact that our mice were heterozygotes as compared to the homozygous ROSAH model mice developed previously[3]. While our mice thus more closely mimic the heterozygous presentation of human ROSAH syndrome patients, the more muted effects of a single copy of human ALPK1[T237M] may have translated to serum chemokine production at levels below the limits of detection for our ELISAs.

Our model animals allowed us to analyze T237M-related changes in ROSAH pathology. In line with the autoinflammatory nature of ROSAH syndrome, our ROSAH mice presented with increased microglial activation and migration from the inner and outer plexiform layer where they normally reside, into the INL and ONL of the retina, potentially thereby exacerbating retinal degeneration[21]. These mice also exhibited astrocyte reactivity that can be indicative of poor retinal ganglion cell health[22]. Importantly, daily oral DF-003 treatment for 10 days was sufficient to suppress T237M-induced microglial infiltration and astrocyte activation. Notably, we detected significant $Il6$ upregulation in the retinas of our hALPK1[T237M] mice. Kozycki et al.[3] noted that ROSAH patients frequently present with elevated plasma IL-6 levels and that tocilizumab improved intraocular inflammation (two of two patients) while also positively impacting other parameters after treatment initiation[3]. As DF-003 treatment (5 mg/kg) reduced $Il6$ upregulation in our ROSAH model mice, this offers further support for its potential clinical utility in ROSAH patients.

A limitation of our study is that ROSAH mice do not show signs of visual acuity loss, and it was therefore not possible to determine if the DF-003-mediated alleviation of ocular inflammation in mice could lead to vision improvement. To our knowledge, neither gene expression nor retinal histology has been investigated in ROSAH patients and thus, we cannot conclude whether the observations made in our model mice also apply to humans. Additionally, it remains to be determined whether DF-003 can reduce ocular inflammation in humans as it does in our model, and if so, whether reduced inflammation in the eye can ultimately lead to the cessation of retinal degeneration in ROSAH patients.

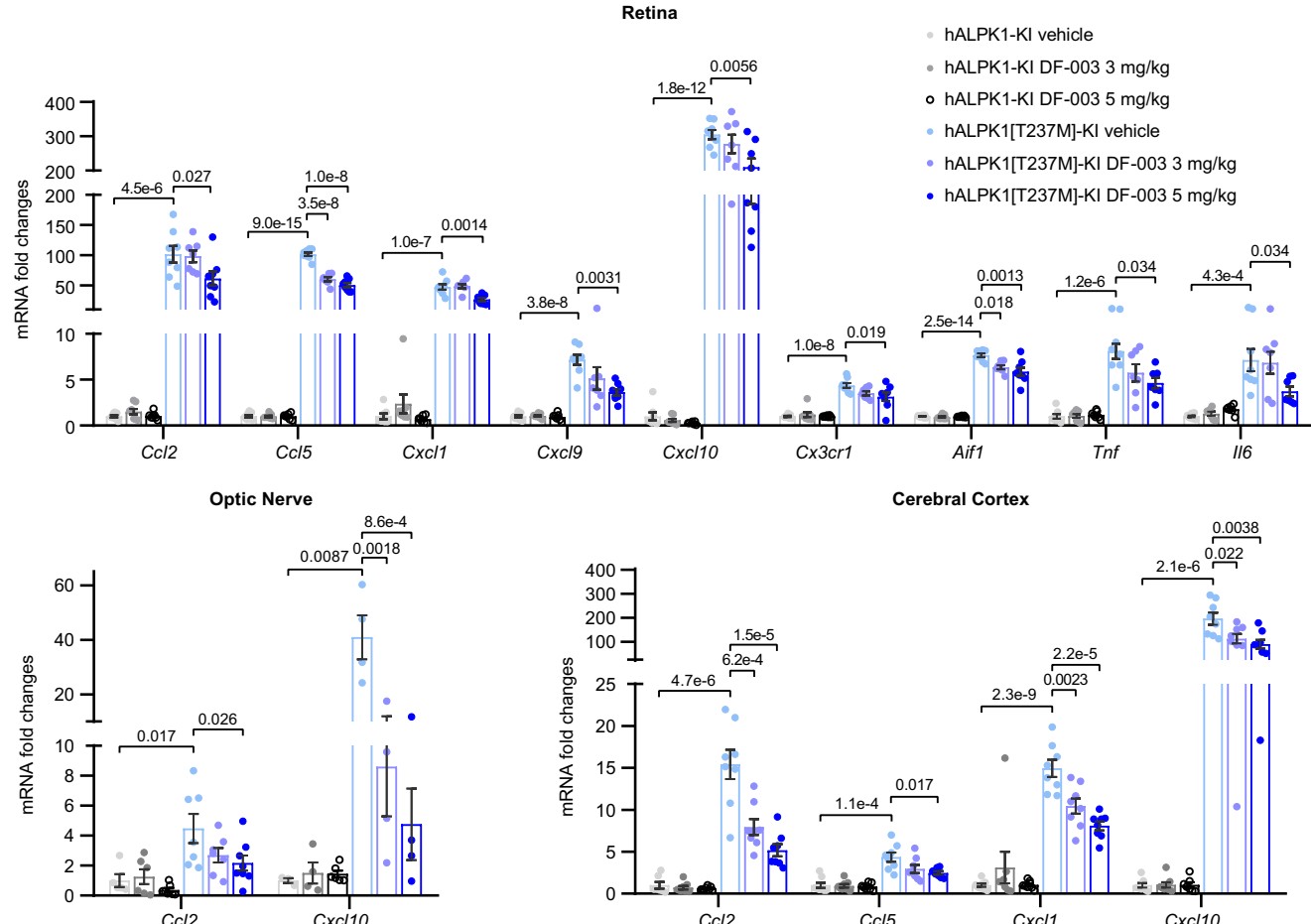

**Fig. 6 | DF-003 suppresses pro-inflammatory gene upregulation in the retina, optic nerve, and brain cortex of ROSAH model mice.** Female hALPK1-KI and hALPK1[T237M]-KI mice were treated orally with DF-003 once per day for 10 days at the indicated doses ($n$ = 8 animals for each group). The following day, animals were euthanized and the mRNA levels of the indicated genes in the retina, optic nerve, and brain cortex samples from these mice were analyzed by qPCR and normalized to levels of *Rpl13*. Data represent means ± SEM for each genetic and treatment group, with each dot representing an organ sample from one animal. In some analyses, group sizes were <8 due to RNA sample loss during in vitro procedures or due to the detection limit. Statistical outliers were identified using Grubb's test. All excluded data points were indicated in the Source Data file. Statistical comparisons between vehicle groups for the two genotypes were made using two-tailed unpaired Student's *t*-tests, and comparisons between DF-003-treated groups to the vehicle group within the same genotype were made with one-way ANOVAs followed by Sidak's post hoc tests. *p*-values <0.05 have been indicated on the charts.

Although cognitive deficits are not a common finding in ROSAH patients, half of patients reportedly experience recurrent headaches, and MRI analyses of some patients have revealed brain abnormalities including age-related premature calcification and white matter abnormalities[3], consistent with potential inflammatory activity in the CNS. While we did not perform MRI-based analyses of the brains in our ROSAH model mice, the observed basal upregulation of chemokines in cortical regions of the brain in our hALPK1[T237M]-KI mice aligns with the human reports and highlights another avenue for future follow-up focused on CNS pathology. Strikingly, DF-003 was able to access the brain and reverse ALPK1[T237M]-induced increases in cortical chemokine expression in our ROSAH mouse model, further supporting its ability to cross the blood-brain barrier and emphasizing the need for further exploration of its effects on potential ROSAH-related CNS changes.

Based on our results, there are several important avenues for future research. Single-cell transcriptomic and proteomic analyses may be used to identify the cell types responsible for the autoinflammatory status of our model mice. Reciprocal adoptive transfer experiments may also be implemented to study the relative contributions of bone marrow-derived immune cells and tissue-resident immune cells in the pathogenesis of ROSAH syndrome. With the discovery of the potent, selective ALPK1 inhibitor DF-003, there is also the potential to further interrogate ALPK1 biology in ROSAH syndrome through ex vivo analyses of patient-derived cells, mouse model experiments, and human clinical trials. We also cannot exclude the possibility of DF-003 having an immunosuppressive effect that could increase the risk of infection. However, in 28 day Good Laboratory Practice (GLP) toxicology studies performed in rats and dogs, no general immunosuppression was observed. Nonetheless, it will be important to monitor for immunosuppression and infection risk in clinical trials.

In conclusion, these data offer strong preclinical support for DF-003 as a potent and targeted inhibitor for the root cause of ROSAH syndrome. The investigational drug DF-003 has completed human clinical testing in a Phase 1 trial (NCT05997641) and is now entering a Phase 1b trial in adult ROSAH patients (NCT06395285). In this Phase 1b trial, outcome measures including safety, pharmacokinetics, pharmacodynamics, eye evaluations, headache, and quality of life will be assessed.

## Methods

### Ethics statement

All animal studies were approved by the Institutional Animal Care and Use Committees (IACUC) at Shanghai Yao Yuan Biotechnology Ltd.,

Zhejiang Yao Yuan Biotechnology Ltd. and Oujiang Laboratory under protocol numbers DF00320231017-001 and DF00320231109-001. All animals received humane care in accordance with the national guidelines for housing and care of laboratory animals (Ministry of Health, Beijing, China).

## Sex as a biological variable
Both male and female mice were used to conduct the experiments reported herein, and no specific or consistent differences were observed as a function of sex.

## Nucleotide sugars
Adenosine-diphosphate-D-glycero-β-D-manno-heptose (ADP-D-Heptose), used in the TR-FRET kinase, radiolabeled kinase, and ADP-Glo kinase assays, as well as uridine-diphosphate-β-D-mannose (UDP-mannose) used in the radiolabeled kinase assay, were synthesized in-house following previously published synthesis routes[23–28]. D-glycero-D-manno-6-fluoro-2,3,4,7-Ac-heptose-1β-S-ADP (DF-006) used in cell-based assays, was also generated in-house[29] and has been characterized previously[30]. Adenosine-diphosphate-L-glycero-β-D-manno-heptose (ADP-L-heptose) used in surface plasmon resonance (SPR) experiments was purchased from InvivoGen (tlrl-adph-l, Hong Kong, China).

## Design, synthesis, and characterization of DF-003
Please see the Supplementary Notes, 1–2 in the Supplementary Information.

## Recombinant protein preparation
Recombinant ALPK1 proteins used for thermal shift assay (TSA), TR-FRET (time-resolved Förster resonance energy transfer)-based kinase assay, radiolabeled kinase assay, and SPR assay, as well as the recombinant ALPK1[T237M] protein used for the radiolabeled kinase assay were purified from Sf9 insect cells. DNA sequences coding for human ALPK1 (NP_079420.3) and ALPK1[T237M] with 3 × Flag-tag (MDYKDHDGDYKDHDIDYKDDDDK) sequences added to the C-terminus of the protein were codon optimized for optimal expression in insect cell lines. The DNA sequence was cloned into the pFAST-BAC vector for the generation of a baculovirus used to transduce insect Sf9 cells to overexpress these recombinant proteins. Recombinant ALPK1 used in a Western blotting-based kinase assay for verification of the TSA screen was purified from human 293-F cells. The same codon-optimized ALPK1 coding sequence, with 3 × Flag tags at the N-terminus, was cloned into the pcDNA3.1 vector. 293 F cells were transiently transfected for 48 h, while in the last 4 h 4% *Salmonella typhimurium* (*s.t.*) lysate was added to the media[31]. Recombinant proteins used for site-directed mutagenesis analysis to validate homology model were purified from expi293 cells without *s.t.* lysate stimulation. Sf9, expi293, or 293-F cells were homogenized in 50 mM Tris, 0.5 M NaCl, 5% glycerol, pH 8.0; proteins were bound to anti-Flag affinity beads, washed, and eluted using 3 × Flag peptides in the same solution. The coding sequence for human TIFA was synthesized and cloned into the pET28a (+) vector, with 6×His-tag or HA-6×His-tag sequences having been added to the 5′ end of the coding sequence. These vectors were used to transform *Escherichia coli*. After lysing the bacteria, recombinant His-tagged and HA-His-tagged TIFA were both purified with a Ni column and were finally dissolved in a buffer containing 50 mM Tris and 150 mM NaCl (pH 8.0) after desalting. His-TEV-tagged Mouse ALPK1 (NP_082084.1) kinase domain aa. 949-1231 (mALPK1-KD) and human ALPK1 N-terminal regulatory domain aa. 1-446 (ALPK1-ND) used for crystallography were purified from *E. coli*. using a similar approach except that the His-tag was cleaved with a TEV protease.

## Crystallography
The mouse kinase domain (mALPK1-KD; amino acids 949-1231) and human N-terminal regulatory domain (ALPK1-ND; amino acids 1-446) were mixed and allowed to form a complex and further purified using size exclusion chromatography and concentrated to 10 mg/ml. A co-crystal was obtained using a microtiter plate format from a well containing a solution of 12% PEG 3350 and 0.1 M cadmium bromide tartrate polyethylene glycol (CBTP), pH 8.2, with 0.1% seed crystal, by hanging drop vapor diffusion. The X-ray diffraction data was collected by Synchrotron (Shanghai Synchrotron Radiation FacilityBeamline BL17U) at the wavelength of 0.979 Å. The data was processed using the programs DENZO and SCALEPACK. The phase was determined by SAD (Single-wavelength Anomalous Dispersion), and automatic model building was performed in PHENIX. The rest of the model was manually built with Coot and refined in PHENIX.

## Homology modeling and molecular docking
Homology modeling of ALPK1 was performed using the Prime module in Schrödinger Maestro Suite 2021-02 following standard knowledge-based model building methods. From the crystal structure of ALPK1 and Dictyostelium myosin II heavy chain kinase A (3PDT), a prototypical member of the atypical alpha-kinase family of kinases with a conserved domain organization similar to ALPK1, the ATP binding pocket was built by identifying domain movement, building a chimera structure and remodeling the activation loop (under the OPLS4 force field). This model was optimized prior to docking using the protein preparation workflow in Schrodinger Maestro Suite 2021-02. Energy minimization of the protein structure was carried out using OPLS4. Molecular dynamics (MD) simulations for the generation of multiple protein conformations were used to address flexibility in the binding site. The model was then refined by induced fit docking (Induced Fit Docking Panel, standard protocol, using the OPLS_2005 force field) to generate grid-box coordinates for docking and validated by chemistry structure-activity relationship (SAR) analyses. The optimized grid was used for docking analysis (Glide module, Standard precision mode, using the OPLS_2005 force field and setting the hydrogen bond interaction of hinge binding as a constraint) of DF-003 and ATP.

## Molecular dynamics simulation of the ALPK1-DF-003 complex
Molecular dynamics (MD) simulation results can provide insights into protein-binding interactions between molecules and proteins. For this study, a 200 ns MD simulation was performed for the ALPK1-DF-003 complex using the Desmond module from Schrodinger Release 2021-02, the optimized potentials for liquid simulation (OPLS4) force field at pH 7.4. Before conducting this MD simulation, the complex was solvated in SPC solvent within an orthorhombic box and counterions (Na$^+$, Cl$^-$) were added to maintain a 0.15 M salt concentration. The simulation was completed under a bar pressure of 1.01325 and a constant temperature of 300 K, with a mainlining recording interval of 200 ps (Supplementary Notes 1, Table 8). The structural stability of DF-003 within the ALPK1-DF-003 complex was assessed using the root mean square deviation (RMSD) of protein and DF-003. The RMSD of the protein was calculated based on atom selection, and changes of the order of 1–3 Å were considered acceptable for a stable structure. The RMSD of the ligand was measured after the protein-ligand complex was first aligned to the protein backbone. If the observed value was not significantly larger than the RMSD of the protein, then the ligand had not diffused away from its initial binding site and the structure was considered stable. The simulation was performed three times under the same conditions to verify the consistency of the observed results.

## High-throughput screening using a thermal shift binding assay
A small molecule library containing ~160,000 compounds (HTS Diversity V2) was prepared by WuXi AppTec (Shanghai, China) in 384

well thin-wall qPCR plates, with 20 nL of 10 mM compound preparations in each well. On each screening plate, 2 columns of wells were loaded with 20 nL of pure DMSO to leave spaces for negative controls (no compounds) and the positive control (AS-252424). Using a Multidrop™ Combi (Thermo Fisher Scientific, Waltham, MA, USA), 3 mixtures were added to each well sequentially and mixed: 5 µl of compound solvent (50 mM Tris, 500 mM NaCl, 2% DMSO, pH 8.0) to compound wells and negative control wells, 3 µl of recombinant full-length ALPK1 kinase purified from Sf9 cells diluted in 50 mM Tris, 500 mM NaCl pH 8.0, and 2 µl of 5× Protein Thermal Shift™ Dye (Thermo Fisher Scientific, 4461146). The final concentration of ALPK1 was 150 µg/mL (~1.1 µM), and the compound concentration was 20 µM. The mixture was spun down at 1000 rpm for 1 min, then immediately applied to a protein melting reaction on a Roche LightCycler® 480 II Real-time PCR system, heating the mixture from 30 °C to 50 °C at the ramp rate of 0.02 °C/s while continuously acquiring the fluorescence signal. The melting temperature ($T_m$) of the ALPK1 protein with or without each compound was calculated using the Protein Thermal Shift™ Analysis Desktop Software v1.2 (Thermo Fisher Scientific). For positive control wells, 40 µM AS-252424 in 5 µl compound solvent was added manually and used as a positive control in the thermal shift screening. AS-252424 was discovered to cause a 3.4 °C increase in $T_m$ in a pilot TSA screening using a kinase inhibitor library (EFEBIO, Shanghai, China). Binding of AS-252424 and 6 other hits from pilot screening to ALPK1 was confirmed in surface plasmon resonance (SPR) experiments (described below) to validate the TSA method. Compounds that caused an increase in $T_m > 1$ °C were considered to be hits and were tested in a Western blotting-based kinase assay and/or a TR-FRET kinase assay (described below) to investigate whether the binding leads to the inhibition of ALPK1 kinase activity and to measure the relative inhibitory potency of each inhibitory compound for the SAR study.

### Surface Plasmon Resonance (SPR)

SPR assays were conducted on a Biacore 8 K (Cytiva). A Series S NTA chip (Cytiva) was first conditioned with 350 mM EDTA for 60 s at 30 µL/min, activated with 0.5 mM NiCl₂ for 60 s at 10 µL/min, and then activated with a 50/50 mixture of 0.04 M EDC and 0.1 M NHS (final concentrations 0.02 M and 0.05 M, respectively) for 420 s at 10 µL/min. Recombinant ALPK1 (purified from Sf9 cells, final concentration 120 µg/mL, 857 nM) in immobilization buffer (10 mM HEPES pH 7.4, 150 mM NaCl, 0.05% Tween-20, 0.5 mM TCEP) was injected at 5 µL/min with a contact time of 1000 s. The chip was then deactivated with 1 M ethanolamine-HCl pH 8.5 for 420 s at 10 µL/min and washed with 350 mM EDTA for 60 s at 30 µL/min to remove Ni²⁺. For analyte testing, flow rates were maintained at 30 µL/min. An ADP-L-heptose 500 nM solution was prepared in running buffer (10 mM HEPES pH 7.4, 150 mM NaCl, 0.05% Tween-20, 0.5 mM TCEP, 1% DMSO). Analyte was 2-fold serially diluted in the same running buffer containing 500 nM ADP-L-heptose, with a maximum final concentration of 1 µM for DF-003 and 50 µM for selected TSA chemical hits. Analyte binding was measured using an A-B-A method, beginning with a 50 s injection of 500 nM ADP-L-heptose ("A") followed by injection of analytes ("B") and dissociation of analytes in B. For $K_D$ measurements, the association time was 80 s and the dissociation time was 120 s. The data collection rate was 10 Hz, and all data were double referenced and solvent corrected. Analysis was performed using Biacore Insight Evaluation software version 5. The 1:1 kinetic binding affinity models were used to fit the data, and all parameters were fitted globally.

### Western blotting-based in vitro kinase assay

Compounds were tested at a single final concentration of 20 µM in an in vitro kinase assay including 30 nM recombinant ALPK1 (purified from 293-F cells stimulated with 4% s.t. lysate 4 h before cell lysis) and 1.6 µM His-tagged TIFA in a 20 µl volume. Recombinant ALPK1

produced in this fashion exhibited kinase activity such that it was able to phosphorylate TIFA without any additional agonist. An in vitro kinase reaction was conducted at 27 °C for 40 min in the following buffer: 20 mM HEPES pH 7.0, 1 mM TCEP, 10 mM MgCl₂, 0.1 mM Na₃VO₄, 0.6 µM BSA, 5% DMSO and 25 µM ATP. Reactions without compound (replaced with pure DMSO) and reactions with neither compound nor ATP were regarded as negative control (0% inhibition) and positive control (100% inhibition), respectively. After these in vitro kinase reactions, the kinase-substrate mix was fractionated by electrophoresis on denaturing SDS-PAGE gels and transferred to Immobilon-P PVDF membranes (Millipore, MA, USA). Membranes were blocked with 5% milk in TBS with 0.1% Triton-X100. Phosphorylated TIFA were detected by incubating the membrane with an anti-phospho-threonine (#9386, Cell Signaling Technology, Danvers, MA, USA) antibody.

### TR-FRET kinase assay

In each well of a 384-well plate, three mixtures were added sequentially: 2 µL of blank control or serially diluted compounds (inhibitory binders identified from the TSA screen or compounds following SAR optimization) in 5% DMSO, 4 µL of substrate/kinase mix (final concentrations of 260 nM recombinant HA-His-tagged TIFA, 2 nM recombinant ALPK1, 50 mM HEPES, 10 mM MgCl₂, 1 mM EGTA, 2 mM DTT, 0.01% (v/v) Tween-20, pH 7.5), and 4 µL of ATP/ADP-D-heptose mix (final concentrations of 6.25 µM ATP and 20 nM ADP-D-heptose). The 10 µL kinase reactions were allowed to proceed for 1.5 h at 27 °C. Kinase reactions were terminated by adding 10 µL of the HTRF KinEASE detection kit buffer (Cisbio, Bedford, MA, USA; 62SDBRDF) containing monoclonal anti-HA-Eu cryptate (Cisbio; 610HAKLB) and monoclonal Anti HA-d2 to each well. TIFA phosphorylation levels were monitored through the detection of oligomerization signals generated upon TIFAsome assembly. Both donor and acceptor antibodies recognize TIFA and will generate a TR-FRET signal when bound to the same TIFA oligomer formed upon TIFA phosphorylation. After incubation at 26 °C for 1 h, fluorescent signals at 620 nm and 665 nm, for europium donor and d2 acceptor fluorescence, respectively, were detected using a Tecan INFINITE NANO+ microplate reader (Tecan, Männedorf, Switzerland). The TR-FRET signals were calculated as the ratio of the acceptor and donor emission signals with the formula below:

Ratio = Signal at 665 nm/ Signal at 620 nm × 10,000

ALPK1 kinase activity was calculated as follows: % Activity = 100% × (TR-FRET signal of test compound - mean TR-FRET signal of no ATP 1% DMSO control) / (mean TR-FRET signal of 1% DMSO control – mean TR-FRET signal of no ATP 1% DMSO control).

In multi-dose assays, compound activity was calculated via logarithmic interpolation. Concentration-response curves were fitted with a four-parameter logistic nonlinear regression model and the $IC_{50}$ values for these compounds were calculated in GraphPad Prism 6. In early studies, eltrombopag and tolcapone were identified as hits from a pilot TSA screen of an FDA-approved drug library and were found to inhibit ALPK1 activity in the Western blotting-based assay described above. Both compounds also abolished the TR-FRET signal with $IC_{50}$ of 7.6 µM and 18 µM, respectively, validating the TR-FRET assay. In one case, the inhibitory potency of DF-003 was tested at increasing ATP concentrations ranging from 0.15625 µM to 25 µM.

### High-throughput screen with TR-FRET kinase assay

To identify additional ALPK1 inhibitor pharmacophores, a second high-throughput screen was conducted using a small molecule library containing ~200,000 compounds (HTS Diversity V3) prepared by WuXi AppTec in 384-well plates, with 10 nL of 10 mM compound preparations in each well. The TR-FRET kinase assay was conducted as described above, except that 4 µL of 2.5% DMSO was first added to each well to dilute the compounds therein, followed by 4 µL of substrate/ kinase mix (final concentrations of 260 nM recombinant HA-His-

tagged TIFA, 2 nM recombinant ALPK1) and 2 μL of ATP/ADP-D-heptose mix (final concentrations of 6.25 μM ATP and 20 nM ADP-D-heptose). Sixteen repeats of negative control [vehicle (1% DMSO) with ATP], 16 repeats of positive control (vehicle without ATP) and 16 repeats of reactions containing 20 μM compound **23** (Supplementary Notes, 1 Table 3) used as positive control were conducted on each screening plate. Compounds that inhibited >40% of ALPK1 kinase activity at a final concentration of 10 μM were considered as hits and applied to subsequent dose-dependent potency analyses and SAR optimization.

### In vitro radiolabeled kinase assays

Assays were conducted by Reaction Biology (Malvern, PA, USA). To measure DF-003 $IC_{50}$ value for ALPK1, recombinant human ALPK1 (0.5 nM), human TIFA (10 μM), and ADP-D-heptose (5 nM) were mixed in a kinase reaction buffer (20 mM HEPES (pH 7.5), 10 mM $MgCl_2$, 1 mM EGTA, 0.01% Brij35, 0.02 mg/mL BSA, 0.1 mM $Na_3VO_4$, 2 mM DTT, 1% DMSO) to which 3-fold dilutions of DF-003 (1 μM - 50.8 pM; prepared in DMSO) were added. After the addition of [$^{33}$P]-ATP (20 μM; specific activity: 0.01 μCi/μL final), the kinase reaction was allowed to proceed at room temperature for 60 min, and samples were then spotted onto P81 ion exchange paper (#3698-915, Whatman). After extensive washing with 0.75% phosphoric acid, the radioactive phosphorylated substrate remaining on the filter paper was measured to quantify ALPK1 kinase activity, which was expressed as the percentage of remaining ALPK1 kinase activity relative to vehicle (DMSO, no DF-003) conditions. Curve fitting and $IC_{50}$ value calculations were performed using GraphPad Prism 4. The same kinase assay was used to measure DF-003 $IC_{50}$ values for ALPK1[T237M], except that ALPK1 was replaced by ALPK1[T237M] and ADP-D-heptose was replaced by 10 μM UDP-mannose. The time-course of TIFA phosphorylation by ALPK1[T237M] agonized by 5 nM ADP-D-heptose or 10 μM UDP-mannose was measured by detecting radioactive phosphorylated substrate on the filter paper at 20, 40, and 60 min after reaction initiation in the same kinase reaction condition without addition of DF-003.

### In vitro ADP-Glo kinase assays

For assessing kinase activity of ALPK1 and its single amino acid substitution mutants, the ADP-Glo™ (Promega, V9101) reagent was used following the manufacturer's instructions, with recombinant human ALPK1, ALPK1-GLU1137A, ALPK1-GLU1137K, or ALPK1-TYR1133A (all used at 2.5 nM), human TIFA (1.7 μM), and ADP-D-heptose (20 nM). The Km of ATP for each enzyme was first determined by measuring the velocity of ADP production under various ATP concentrations ranging from 2 μM–500 μM. The $IC_{50}$ of DF-003 for each enzyme was determined by adding half-log10 serial dilutions of DF-003 (5 μM - 158 pM; prepared in DMSO) with ATP added to the kinase reaction at the Km concentration (16 μM for ALPK1, 113 μM for ALPK1-GLU1137A, 33 μM for ALPK1-GLU1137K, and 49 μM for ALPK1-TYR1133A). Kinase reactions were conducted at 28 °C for 40 min. Curve fitting and Km and $IC_{50}$ value calculations were performed using GraphPad Prism 4.

### HotSpot kinase assay

The inhibitory effects of DF-003 on the activity of 394 human kinases were measured by Reaction Biology using the previously described HotSpot™ Kinase Assay[32]. The general experimental approach was the same as the in vitro kinase assay detailed above, with an extended 120-min reaction time, an ATP concentration of 10 μM, and a single tested DF-003 dose (10 μM). The percentage of kinase activity was calculated for each kinase with DMSO and DF-003 treatment. The dose-dependent inhibitory activity of DF-003 against the top 3 non-ALPK1 human kinases that were most strongly inhibited by 10 μM DF-003 was further determined in kinase reaction assays using a 10-dose dilution series of DF-003 (20 μM–0.763 nM), with all other experimental conditions being the same as above. Additionally, a 10-dose dilution series

of staurosporine and other enzyme-appropriate compounds were included as positive control inhibitors.

### Cell preparation and treatment

THP-1 cells (Cell Bank of The Chinese Academy of Sciences, Beijing, China) were routinely cultured in RPMI-1640 (Hyclone, Logan, UT, USA) containing 10% heat-inactivated fetal bovine serum (FBS, Hyclone), 0.05 mM 2-mercaptoethanol, 1% penicillin (100 U/mL), and streptomycin (100 μg/mL) (Gibco, Waltham, MA, USA) in a humidified 5% $CO_2$ incubator at 37 °C. To induce macrophage-like differentiation, these cells were seeded in 24-well plates ($4\times10^5$/well) and treated for 48 h with phorbol myristate acetate (PMA; 50 ng/mL). Media was then exchanged for fresh complete medium without PMA and pretreated with a range of DF-003 concentrations for 2 h (0.3 nM – 1000 nM), after which they were stimulated for an additional 4 h using 5 nM DF-006. Cells were then collected for qPCR to analyze the expression of *TNF* and *CXCL8* (See below). Samples were analyzed in quadruplicate, and $IC_{50}$ values for the inhibition of ALPK1 agonist-induced cytokine upregulation were calculated by plotting relative mRNA expression against DF-003 concentration for each gene of interest using Graph-Pad Prism 6.

HEK-293 cells were purchased from the Cell Bank of The Chinese Academy of Sciences and routinely cultured in DMEM/High-Glucose (Hyclone). For TIFAsome analyses, the coding sequence for the GFP-TIFA fusion protein was cloned into the pcDNA3.1 plasmid. HEK-293 cells cultured on coverslips were transiently transfected with the construct, treated with either 1 μM or 200 nM DF-003 (or DMSO as a vehicle control), followed by 1 μM DF-006 stimulation for 1 h. Each condition was conducted in sextuplicate. Cells were fixed with 4% paraformaldehyde and stained with DAPI (Shanghai Zhenghuang, Shanghai, China; C1005). Fluorescent images were acquired using a Motic upright fluorescent microscope (PA53 BIO FS6). For each coverslip, five 2560 μm × 1760 μm fields were imaged and analyzed by a researcher blinded to treatment conditions, calculating the percentage of TIFAsome-positive cells among all positive cells exhibiting green fluorescence.

To obtain HEK-293 cells stably overexpressing ALPK1 or ALPK1[T237M], codon-optimized DNA coding sequencing for Flag-tagged human ALPK1 and ALPK1[T237M] were inserted into the pcDNA3.1 vector. Plasmids were transfected into HEK-293 cells using Lipofectamine 2000 as per the manufacturer's protocol. At 48 h post-transfection, G418 (#A1720, Sigma-Aldrich, St. Louis, MO, USA) was used to select resistant transformed cells, and single-cell clones were prepared through a limiting dilution-based approach. Western blotting (See Below) was used to assess Flag-tagged ALPK1 and ALPK1[T237M] overexpression. Cells were treated in triplicate with a range of DF-003 concentrations (0.64 nM – 20 μM) for a total of 30 h, during which media was refreshed with equivalent DF-003 doses after 24 h. Cells were then harvested to analyze the expression of *CXCL10, TNF*, and *CXCL8* by qPCR (See Below). CXCL8 concentrations in supernatants collected from these cells were analyzed with a Human CXCL8 ELISA Set (#555244, BD Biosciences, Franklin Lakes, NJ, USA) based on the manufacturer's instructions, and results were analyzed with a micro-plate reader (Multiskan FC, Thermo Fisher Scientific).

To prepare stable NF-κB reporter cells, HEK-293 cells were transfected with an NF-κB dual reporter construct (a kind gift from Dr. Lei Sun of Fudan University). The construct contains an NF-κB responsive element (5′-GGGAATTTCCGGGAATTTCCGGGAATTTCCGGGAATTT CCGGGAATTTCCGGGAATTTCCGGGAATTTCCGGGAATTTCC-3′), a minimized CMV promoter, followed by coding sequences for secreted alkaline phosphatase (SEAP) and firefly luciferase linked by a 2 A peptide. G418 (#A1720, Sigma-Aldrich) was used to select resistant transformed cells, and single-cell clones were prepared through a limiting dilution-based approach. Positive clones were confirmed by increased alkaline phosphatase activity and luciferase activity after stimulating

cells with DF-006. To measure increased NF-κB signal caused by ALPK1[T237M] and DF-003's inhibitory effect, reporter cell clones were transfected with ALPK1 or ALPK1[T237M] expression vectors. Four hours after transfection, cells were treated with serially diluted DF-003 for an additional 48 h Intracellular luciferase activity was measured using Luciferase Reporter Gene Assay Kit (Beyotime, Jiangsu, China; RG027) per the manufacturer's instructions.

## Quantitative real-time PCR
RNA was extracted from harvested tissue and cell samples using the TRI reagent (T9424, Sigma-Aldrich). The HiScript Q-RT SuperMix (Vazyme, Nanjing, China) was then used to synthesize cDNA. Real-time PCR was carried out using a QuantStudio™ 5 Real-Time PCR System (Applied Biosystems) with the AceQ qPCR SYBR Green Master Mix Kit (Vazyme). All qPCR assays were run in 384-well thin-wall plates with a total reaction volume of 10 μL, including 2 μL of cDNA (20 ng), 5 μL of 2x AceQ qPCR SYBR Green Master Mix, 0.2 μL of ROX Reference dye (50x), 0.1 μL of each primer (F + R, 10 μM of each), and 2.6 μL of ddH₂O. Thermocycler settings included an initial 5 min at 95 °C followed by 40 cycles of 95 °C for 10 s and 60 °C for 30 s. Relative gene expression was analyzed using the $2^{-\Delta\Delta Ct}$ method, and *GAPDH* served as a normalization control for human cell lines and *Rpl13* for murine tissues. Data are presented as the fold-change in expression relative to vehicle control unless otherwise noted. Primers used for these analyses are presented in Supplementary Table 3.

## Pharmacokinetic analyses
Mice were housed in a specific pathogen-free (SPF) animal facility at Drug Farm (Shanghai, China) in individually ventilated cages on a 12 h light/dark cycle with a bedding of wood shavings and *ad libitum* access to rodent chow. The facility was maintained at a temperature of 20 °C to 26 °C with humidity between 40% and 70%. Six 8 week-old male C57BL/6J mice from Zhejiang Vital River Laboratory Animal Technology Co., Ltd. (Zhejiang, China) were randomly assigned to 2 groups (3 animals/group). Animals were administered DF-003 by oral gavage once daily at 3 and 5 mg/kg for 10 consecutive days. DF-003 were dissolved in 0.1% (w/v) methyl cellulose (MC, 1500 cP) + 0.2% (v/v) Tween 80 in purified water. The oral dosing volume for all animals was 10 mL/kg body weight. Drinking water and certified rodent diet were available to animals *ad libitum*, except that animals were fasted 16 h prior to the last administration until 4 h after the last dosing. Blood samples at 0 (right before the last dosing), 0.5, 1, 2, 4, 6, 8, 12, and 24 h after the 10ᵗʰ dosing were collected from the tail vein and transferred into tubes containing K₂-EDTA. The tubes were gently inverted several times to ensure mixing and immediately placed on wet ice. Plasma was obtained by centrifugation at 3200 × g at 4 °C for 10 min and stored at ≤ -60 °C until subsequent analysis. Twenty-four hours after the 10th dosing, the animals were euthanized by CO₂ inhalation. The eyes were harvested, and the retinas and optic nerves were separated. In addition, a piece of cerebral cortex tissue (~50 μg) from each mouse was snap-frozen in liquid nitrogen. The concentrations of DF-003 in the plasma, retina, and cerebral cortex of each mouse were measured by Wuhan Haipu Biomed Inc. (Wuhan, China) using an LC-MS/MS approach. The concentration of DF-003 in the optic nerve was also measured using this same strategy, combining all 6 optic nerves from the 3 mice in each dosing group as a single sample.

## Mouse model of ROSAH syndrome
*Alpk1⁻/⁻* mice (with removal of exon 13 to abolish the kinase activity of endogenous mouse ALPK1) were generated previously[12]. To prepare ROSAH model mice using CRISPR, a guide RNA (gRNA) (5'-AGTGAG-GACCAGCGGTGCAGAGG-3') targeting exon 2 of the mouse *Alpk1* gene (NCBI Gene ID: 71481) was generated. Targeting recombination vectors containing the following elements were prepared: (1) a 1.5 kb 5' arm homologous to the genomic sequence upstream of the start codon of

mouse *Alpk1* (including ATG); (2) the human *ALPK1* coding sequence (encoding wild-type ALPK1 for the *mAlpk1ʰALPK1* allele or ALPK1[T237M] for the *mAlpk1ʰALPK1T237M* allele, the latter of which contains the same C > G nucleotide substitution as in human ROSAH syndrome patients) with a poly-A tail; (3) a 1.4 kb 3' arm homologous to the genomic sequence downstream of the start codon for mouse *Alpk1* (excluding ATG). In vitro transcribed gRNA and Cas9 mRNA, together with targeting recombination vectors, were co-microinjected into C57BL/6N *Alpk1⁻/⁻* mice-derived fertilized eggs. Founders with the *hALPK1* CDS inserted into the mouse *Alpk1* gene were screened via Southern blotting and further confirmed by PCR and Sanger sequencing and were backcrossed with *Alpk1⁻/⁻* mice.

Mice bearing the abovementioned alleles were intercrossed to generate founder *mAlpk1ʰALPK1/hALPK1T237M* mice on the C57BL/6N background (*Crb1ʳᵈ⁸/ʳᵈ⁸*). These mice were backcrossed with wild-type C57BL/6J (*Crb1⁺/⁺*) mice (purchased from Zhejiang Vital River Laboratory Animal Technology Co., Ltd.) twice to respectively obtain *mAlpk1⁺/ʰALPK1*; *Crb1⁺/⁺* and *mAlpk1⁺/hALPK1T237M*; *Crb1⁺/⁺* mice. These lines were intercrossed to generate *mAlpk1ʰALPK1/hALPK1*; *Crb1⁺/⁺* (designated as hALPK1-KI) and *mAlpk1ʰALPK1/hALPK1T237M*; *Crb1⁺/⁺* (designated as hALPK1[T237M]-KI) mouse lines. Both lines exhibit loss-of-function for mouse Alpk1. Sixteen to seventeen-week-old hALPK1-KI and hALPK1[T237M]-KI littermates from the mating between these two lines were used for testing DF-003 efficacy in vivo. Our mice included the introduction of the wild-type *Crb1* gene from C57BL/6J mice to address the rd8 mutation in this gene present in the C57BL/6N subline that can confound efforts to analyze ocular phenotypes[33].

## In vivo efficacy studies
To test the in vivo efficacy of DF-003 in ROSAH model mice, female hALPK1-KI and hALPK1[T237M]-KI mice (16-17 weeks of age; n = 24/genotype) were each randomized into vehicle (n = 8/genotype), DF-003 3 mg/kg (n = 8/genotype), and DF-003 5 mg/kg (n = 8/genotype) treatment groups, ensuring that there were no differences in starting body weight among groups. Mice were dosed orally with DF-003 (3 or 5 mg/kg) or vehicle control (0.1% MC) once per day for 10 total doses at a dosage volume of 10 mL/kg body weight. The status and body weights of all mice were monitored daily throughout this study. Twenty-four hours after the final dose, one retina, one optic nerve, and brain cortex samples were harvested from each mouse and were immediately homogenized in the TRI reagent (Sigma-Aldrich) and stored at -80 °C for analyses of gene expression. One eyeball from each mouse was fixed with 4% paraformaldehyde, and the retina was dissected and embedded in O.C.T. compound (Sakura Finnetek, Torrance, CA, USA; 4583) followed by storage at -80 °C.

## Histological, immunofluorescent and immunohistochemical staining
Mouse retina samples were fixed in 4% paraformaldehyde (PFA), mounted in O.C.T embedding compound, and frozen at -20 °C to -80 °C, after which they were cut into 10 μm transverse sections near the center of the retina (through the optic papilla).

For immunofluorescent and immunohistochemical labeling, frozen sections were air-dried and then blocked using goat serum diluted in PBS containing 0.1% Triton X-100. The microglia were detected by immunofluorescence using primary rabbit anti-IBA1 (Wako, Richmond, VA, USA; 019-19741) and secondary goat anti-rabbit IgG (H + L) Cross-Adsorbed, Alexa Fluor™ 488 (Invitrogen, Carlsbad, CA, USA; A-11008). Nuclei were counterstained with DAPI (Shanghai Zhenghuang, Shanghai, China; C1005). The astrocyte activation marker GFAP was detected by immunohistochemistry with rabbit anti-GFAP (Abcam; ab68428) and secondary donkey anti-rabbit IgG H&L (HRP) (Abcam; ab6802). ImmPACT® DAB Peroxidase (HRP) Substrate (Vector Labs, Burlingame, CA, USA) was used for staining, followed by diaminobenzidine (DAB) coloration according to routine

immunohistochemistry procedures, yielding a brown signal corresponding to GFAP positivity. Cell nuclei were counterstained with hematoxylin (Baso, Zhuhai, China) and dyed blue.

For hematoxylin and eosin (H&E) staining, cells were initially stained with hematoxylin, followed by differentiation and eosin staining. Images were acquired using a Motic upright fluorescent microscope (PA53 BIO FS6) and analyzed in the CaseViewer software (Version 3.3; 3DHISTECH Ltd., Budapest, Hungary).

For quantification, the Lasso tool was used to draw a curve aligned with the nerve fiber layer of the retina and measured as the length of the retina. The GFAP-positive nerve fiber layer length was measured, summed, and normalized to the overall length of the retina. Total IBA1-positive cells were counted in the whole outer nuclear layer and inner nuclear layer areas of analyzed retinal sections and also normalized to the length of the retina.

### Anhidrosis test
Testing for reduced sweating in experimental mice was performed as described in a previous study[20]. A hind paw of a male hALPK1-KI or hALPK1[T237M]-KI mouse was painted with 2% (w/v) iodine/alcohol solution and allowed to dry. The surface was then painted again with 1:1 starch-castor oil 1:1 (w:v). Purple spots formed by sweating were observed 3–10 min thereafter.

### Assessment of visual acuity
The optomotor response (OMR) test was conducted to assess the visual acuity of unrestrained mice, following the protocol described previously[34]. Mice were placed on a central platform within the qOMR system (PhenoSys, Berlin, Germany), which was enclosed by four screens that were positioned on each wall of a square box. During the test, visual stimulation was presented on the screens. The spatial frequency of the visual pattern varied across a range of frequencies from 0.1 to 0.5 cycles per degree, with increments of 0.05 cycles per degree. For each trial, the visual pattern moved at a constant speed of 12 degrees per second for a duration of 60 s. Throughout this process, the qOMR system automatically tracked and analyzed the head movements of the mice as they attempted to follow the movement patterns. After the test, the system evaluated the accuracy of the mice's tracking behaviors, categorizing them as either correct or incorrect. These data were then comprehensively analyzed to derive the crucial OMR index and determine the visual threshold, as described previously[34]. Initially, we extracted the peak value of the OMR index from the fitted curves. Subsequently, we subtracted 1 from this maximum value and calculated one-quarter of the resulting difference to establish a threshold point. Finally, we aligned this calculated threshold point with the OMR index curve obtained from the experiment to identify the corresponding spatial frequency value. This spatial frequency value represented the visual threshold of the mice, indicating the maximum spatial frequency stimulus that they could accurately track.

### Optical coherence tomography (OCT)
OCT images were obtained using the OPTOPROBE system (Optoprobe, Glamorgan, UK). Mice were anesthetized by intraperitoneal injection of 1% pentobarbital sodium (0.0075 mL/g), and their pupils were dilated with compound tropicamide eye drops. Cross-sectional images of the retina, centering around the optic nerve head, were then captured.

### Fundus imaging
Mice were anesthetized by intraperitoneal injection of 1% pentobarbital sodium and their pupils dilated as described above. The fundus was imaged using the Eyemera fundus camera (IIScience, Busan, South Korea).

### Serum cytokine analyses
Female hALPK1-KI and hALPK1[T237M]-KI mice were treated orally with vehicle or DF-003 (5 mg/kg) once per day for 10 days. Twenty-four hours after the last dose, the indicated plasma cytokine and chemokine levels were measured using the ProcartaPlex Kit (eBioscience, Thermo Fisher Scientific) as directed.

### Statistical analyses
Unless otherwise noted, data are presented as means ± standard error of the mean (SEM). Statistical outliers were detected using a Grubb's test and excluded from further analysis and clearly indicated in Source Data files. Data were compared using independent sample $t$-tests or one-way ANOVAs, as appropriate, and the specific statistical tests used are indicated in the corresponding figure legends. Animal numbers are as noted, and each point in the individual figures corresponds to one animal. A $p$-value $< 0.05$ served as the significance threshold.

### Reporting summary
Further information on research design is available in the Nature Portfolio Reporting Summary linked to this article.

## Data availability
The crystal structure of DF-003 has been deposited in the Cambridge Crystallographic Data Centre under deposition number CCDC 2403779. Copies of the data can be obtained free of charge via https://www.ccdc.cam.ac.uk/structures/. Macromolecular structural data have been deposited in the Worldwide Protein Data Bank Deposition ID 9J4P. Input, parameter, and output files of molecular docking and molecular dynamic simulation analyses are provided in the Supplementary Data 1. The remaining data are available within the Article, Supplementary data files or the Source Data file provided with this manuscript. Source data are provided with this paper.

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

## Acknowledgements

We thank Dr. Jiawei Wang (Tsinghua University), Dr. Yu Zhang (University of Chinese Academy of Sciences), Drs. Thomas Henry and Yvan Jamilloux (CIRI-Centre International de Recherche en Infectiologie), Dr. Zhijian James Chen (University of Texas Southwestern Medical Center), and Dr. Fei Lan (Fudan University) for helpful discussions. We thank Xiaodong Chen for help in the synthesis of ADP-D-Heptose. We thank Junwei Wang, Aimaier Tuerdi, Qiang He, Mingxin Wu, Wei Lu, Yue Wang, Yuxuan Qiu, and Zhenzhen Dong from Shanghai Yao Yuan Biotechnology Ltd and Yuxiang Liu, Yue Li, Biao Xu, and Chaojie Chen from Zhejiang Yao Yuan Biotechnology for experimental assistance. We thank Dr. Ryan Molony for assistance with manuscript preparation.

## Author contributions

T.X. and C.X. conceived the study. J.F., C.X., and H.L. designed the experiments and interpreted the data. D.L., X.W., Z.Z., and W.W. designed and synthesized the chemical entity DF-003. J.F., Z.M., C.Y., S.Z., L.C., S.C., Y.L., L.R., J.C., F.C., X.D., and B.R. performed experiments. Z.C. and J.Z. synthesized UDP-mannose. H.D., Y.P., J.K., R.L., J.Y. and H.S. designed experiments. L.M. and V.M. critically revised the manuscript. J.F., D.L., H.L., and C.X. wrote the manuscript.

## Competing interests

Jieqing Fan, Danyang Liu, Zhu Ming, Chunyu Yan, Yanfang Pan, Huaixin Dang, Xiong Wei, Zhengle Zhao, Wenzhi Wang, Zhenjie Chen, Junlin Zhou, Shuai Zhang, Tong-Ruei R Li, Lawrence Melvin, Jeysen Yogaratnam, Henri Lichenstein, Tian Xu, and Cong Xu own stock in Drug Farm Inc., which funded this work and is the parent company of Shanghai Yao Yuan Biotechnology Ltd, Zhejiang Yao Yuan Biotechnology Ltd, and Drug Farm USA LLC. Vinit Mahajan is a consultant for Drug Farm. Danyang Liu, Cong Xu, Lawrence Melvin, Wei Xiong, Tongruei Raymond Li, Jieqing Fan, Yanfang Pan, Huaixin Dang, Henri Lichenstein and Tian Xu are inventors on two international patent applications filed by Shanghai Yao Yuan Biotechnology Ltd. (PCT/CN2021/119801, filed on September 23 2021, and #19/036,564, filed on January 24, 2025), which are based on the work described in the article. The remaining authors declare no competing interests.
