## [Transparent Peer Review file · Nature Communications]

Discovery of a Selective Alpha-Kinase 1 Inhibitor for the Rare Genetic Disease, ROSAH Syndrome

Corresponding Author: Dr Cong Xu

Version 0:

Reviewer comments:

Reviewer #1

(Remarks to the Author)

NCOMMS-24-46796-T

Jieqing Fan et al., Discovery of a First-in-Class Selective Alpha-Kinase 1 Inhibitor for the Rare Genetic Disease, ROSAH Syndrome

In this manuscript Jieqing Fan and colleagues describe their discovery of DF-003, a highly selective inhibitor of alpha-kinase 1 (ALPK1), and present data supporting the possible use of this agent in the treatment of ROSAH (retinal dystrophy, optic nerve edema, splenomegaly, anhidrosis, headache) syndrome, a disabling human disease caused by gain-of-function mutations in the ALPK1 gene.

Although I am not a pharmacologist, I did find the data on the potency and specificity of DF-003 to be compelling. DF-003 inhibited ALPK1 kinase activity in vitro at nanomolar concentrations, while a screen of 394 other human kinases demonstrated only one other kinase that was inhibited even at micromolar concentrations. Overexpression assays demonstrated that DF-003 inhibited target gene expression induced by the ROSAH ALPK1 [T237M] variant, and the comparison of biochemical with overexpression assays suggests that endogenous nucleotide sugars are required for the T237M gain-of-function effect, as recently suggested by Snelling and colleagues.

To investigate the possible use of DF-003 in ROSAH syndrome, the investigators developed a T237M knock-in mouse model. This model differs from the model described by Kozycki and colleagues in at least two ways:

- 1) The knock-in described by Kozycki et al. involved editing the endogenous mouse *Alpk1* gene, while the knock-in presented in the current paper entailed inserting either the wild type or mutant version of the human coding sequence into the mouse genome;
- 2) The knock-in described by Kozycki et al. was homozygous for the T237M variant, while the knock-in presented in the current paper is heterozygous.

It is interesting to note that, upon sophisticated ophthalmologic examination, neither knock-in mouse demonstrated a loss in visual acuity, at least over approximately one year of observation. It should be noted that many humans with ROSAH syndrome become completely blind, and thus the phenotype is not a subtle one, although noticeable vision loss does not occur until 5-7 years of age, which is beyond the lifespan of a mouse. Fan and colleagues were able to identify histologic changes in the retinas of ROSAH knock-in mice, and increased expression of inflammation-related genes in the retina, optic nerve, and cerebral cortex. These analyses were not performed by Kozycki and colleagues. Kozycki and colleagues found elevations in serum cytokines and chemokines comparable to ROSAH patients, while Fan and colleagues did not observe increased cytokine/chemokine levels in the blood, except for CXCL9. These differences are well within what one might expect, given the differences in the way that the two different knock-ins were generated and the differences in what the two different groups assayed. Although the current paper cannot (and does not) claim to have generated a mouse model that precisely mirrors the human phenotype, the mouse model described in this manuscript exhibits sufficient ocular pathology that responds to DF-003 treatment to warrant further investigation of DF-003 in human ROSAH syndrome.

While this manuscript has a number of strengths, there are areas in which it could be improved;

- 1) Although the authors have made a reasonable case from their knock-in mice that DF-003 warrants investigation in ROSAH syndrome, given the role of ALPK1 in sensing microbial PAMPs, it would probably be worthwhile to assess the effects of an inhibitor like DF-003 in the host response to selected bacterial infections in knock-in mice. It is conceivable that,

even though ROSAH mutations are GOF, there could be an immunosuppressive effect at doses of DF-003 that might overcome the GOF effect;

2) It would be worthwhile for the authors to discuss, all in one place, the comparison of their knock-in mouse data with the data from Kozycki and colleagues;

3) The authors are to be commended for their histologic studies of the retinas of knock-in mice. Are there any comparable data from patients with ROSAH syndrome, especially during the time interval when they are actively losing their vision? Are there any data in ROSAH patients comparable to the gene expression data in the eyes, optic nerves, and brains of the knock-in mice?

4) Given the fact that ALPK1 is a relatively newly recognized kinase, the authors may wish to comment on the fact that DF-003 could be used to interrogate the biology of ALPK1 either in mice or in human subjects who receive the agent as part of a clinical trial;

5) The authors may wish to comment on possible reciprocal adoptive transfer experiments in the T237M and wild-type knock-in mice to determine the role of bone marrow-derived cells in the pathogenesis of ROSAH syndrome;

6) Both in the Introduction and in the Discussion, the authors allude to the inconsistent efficacy of cytokine inhibitors in preventing vision loss in ROSAH syndrome. While, as a class, it is true that cytokine inhibitors show inconsistent effects, there are promising early data that tocilizumab may have a beneficial effect (as noted by the authors). However, at this point there are just not enough data at the level of individual cytokine inhibitors to make any definitive positive or negative pronouncement. These points should be clarified in the manuscript;

7) The authors should consider citing the recent Science paper by Yue Tang and colleagues regarding the coevolution of D-manno-heptoses and ALPK1s, since it provides a broader context for the current manuscript;

8) On page 5 of the manuscript, I believe that on lines 16 and 19 'ALPK1-KD' should be 'mAlpk1-KD'.

Reviewer #2

(Remarks to the Author)

This manuscript by Fan et al reports a small molecule ALPK1 inhibitor as a potential drug candidate for ROSAH syndrome. The absence of approved drugs for the treatment of ROSAH renders this study a useful guide for future research efforts in this direction. The authors designed a nanomolar inhibitor (DF-003) with an adequate pharmacokinetic profile for oral dosing. In addition, the research team performed an extensive panel of screening (in vitro and in vivo) to demonstrate the therapeutic potential of the new compound. However, there are several drawbacks to this work as detailed below that would justify not moving forward with this manuscript for publication in Nature Communications:

1. The design strategy of DF-003 and the details provided on the 2 pharmacophores being combined are insufficient.
2. There is no co-crystal structure reported to elucidate the binding mode of DF-003.
3. The docking studies performed to study the potential binding mode of DF-003 are minimal.
4. Site-directed mutagenesis was not performed to verify the key binding residues predicted to interact with DF-003
5. Since the authors used thermal shift assay to identify hits, I expected extensive MST, SPR, and ITC data to validate the binding of the compound to the target.
6. No sufficient evidence for cellular target engagement of DF-003 is provided.
7. Details regarding the thermal shift assay and TR-FRET (assay development, miniaturization, controls used) are minimal and insufficient to reproduce the work.
8. A med chem campaign focused on the optimization of DF-003 was not pursued.
- 9.

Reviewer #3

(Remarks to the Author)

This is an interesting study that has generated new inhibitors of the ALPK1 protein. My review focusses on the protein structure and docking aspects of the study.

The crystal structure of ALPK1/ADPH is of good quality overall, with geometry and refinement statistics consistent with a 2.2Å resolution. Crucially, the binding pose of the ligand is well-evidenced by clear electron density, and it has good geometry. The kinase part of the structure, and the interface between the two domains are novel findings. The authors could increase the impact of their paper for molecular/structural biologists in the kinase field with a small amount of extra effort to include a supplementary figure showing:

- i) a comparison of the kinase domain structure with the most similar known kinase (i.e. PDB 3PDT)
- ii) details of the molecular interactions at the interface between the N-terminal domain and kinase domain.

The binding mode of DF-003 and its precursors was investigated using molecular docking. The binding site overlaps with that of ATP, consistent with the biochemical data. However, there are a few issues with the modelling that must be addressed:

- i) More information on the homology modelling and molecular docking should be provided (what software was used, what were the key parameters/settings used in the software, what metrics were used to evaluate the reliability of the homology model, are there any alternative binding modes of DF-003 that generated a similar docking score, say within 1 kcal/mole?)

ii) The docking score is comparable with that of ATP, but how does the binding affinity of ATP compare with that of DF-003? If there is a large difference in their affinities, then the similarity of the docking score suggests the method is not fully reliable.

iii) The overall binding mode is reasonable, but some details of the molecular recognition are surprising. Specifically, there are no contacts between the protein and the difluorophenyl group or the ethynyl group. This may reflect small errors in the docking (which may or may not have accommodated induced fit) or possibly an incorrect binding pose. The authors can address this point by explaining how the model is supported by compound SAR (mentioned in the text but no data shown) and by carrying out further docking/molecular dynamics with DF-003 to investigate whether additional, potentially transient interactions are made.

Version 1:

Reviewer comments:

Reviewer #1

(Remarks to the Author)

I am satisfied with the authors' response to my comments, with one small exception. I think that in response to my comment #7, the authors should insert 'across multiple kingdoms' after '(PAMPs)', thus:

Recently, ALPK1 was also shown to be activated by similar pathogen-associated molecular patterns (PAMPs) across multiple kingdoms, including . . .

Reviewer #2

(Remarks to the Author)

The authors have done considerable efforts in addressing the reviewers' comments. However, the points below would need further consideration:

1. SPR analysis was not performed for DF-003
2. SPR sensograms need to show the impact of the multiple doses used to calculate the K_d rather than a single-dose.
3. The authors make a point that site-directed mutagenesis are not essential to validate the preclinical efficacy for DF-003. However, these studies will validate the computational predictions by the authors, which is a major part of the manuscript.

Reviewer #3

(Remarks to the Author)

The authors have addressed all of my initial concerns. I find the new text describing the SAR and methodology compelling.

Version 2:

Reviewer comments:

Reviewer #2

(Remarks to the Author)

The authors have addressed all my major comments properly. I do not have additional comments and find the manuscript to be ready for publication.

Reviewer 1:

In this manuscript Jieqing Fan and colleagues describe their discovery of DF-003, a highly selective inhibitor of alpha-kinase 1 (ALPK1), and present data supporting the possible use of this agent in the treatment of ROSAH (retinal dystrophy, optic nerve edema, splenomegaly, anhidrosis, headache) syndrome, a disabling human disease caused by gain-of-function mutations in the ALPK1 gene.

Although I am not a pharmacologist, I did find the data on the potency and specificity of DF-003 to be compelling. DF-003 inhibited ALPK1 kinase activity in vitro at nanomolar concentrations, while a screen of 394 other human kinases demonstrated only one other kinase that was inhibited even at micromolar concentrations. Overexpression assays demonstrated that DF-003 inhibited target gene expression induced by the ROSAH ALPK1 [T237M] variant, and the comparison of biochemical with overexpression assays suggests that endogenous nucleotide sugars are required for the T237M gain-of-function effect, as recently suggested by Snelling and colleagues.

While this manuscript has a number of strengths, there are areas in which it could be improved;

1) Although the authors have made a reasonable case from their knock-in mice that DF-003 warrants investigation in ROSAH syndrome, given the role of ALPK1 in sensing microbial PAMPs, it would probably be worthwhile to assess the effects of an inhibitor like DF-003 in the host response to selected bacterial infections in knock-in mice. It is conceivable that, even though ROSAH mutations are GOF, there could be an immunosuppressive effect at doses of DF-003 that might overcome the GOF effect;

We thank the reviewer for raising this important point and agree that we cannot exclude the possibility of DF-003 having an immunosuppressive effect. This is a potential challenge faced by any ALPK1 inhibitor therapy. However, in 28-day Good Laboratory Practice (GLP) toxicology studies performed in rats and dogs as a component of our filed IND with the United States FDA, no immunosuppression was observed. We recognize the interest in assessing whether DF-003 affects the host response to selected bacterial infections, but the execution of these experiments is beyond the scope of this paper. We plan to remain acutely aware of the potential immunosuppressive effects of DF-003 and will monitor very closely for this possibility in the context

of longer-term GLP toxicology experiments and in our DF-003 clinical trials. We have updated our Discussion to recognize this important point raised by the reviewer as follows:

“We also cannot exclude the possibility of DF-003 having an immunosuppressive effect that could increase the risk of infection. However, in 28-day Good Laboratory Practice (GLP) toxicology studies performed in rats and dogs, no general immunosuppression was observed. Nonetheless, it will be important to monitor for immunosuppression and infection risk in clinical trials.”

2) It would be worthwhile for the authors to discuss, all in one place, the comparison of their knock-in mouse data with the data from Kozycki and colleagues;

We thank the reviewer for this suggestion, given its importance for comparisons of our study and those from Kozycki et al. We have sought to clarify these points in our revised Discussion as follows:

“Kozycki et al. also previously generated knock-in mice bearing the p.T237M mutation in the murine *Alpk1* gene³, and while their mice exhibited elevated serum concentrations of chemokines including CCL2, CXCL1, and CXCL10, no changes in spleen size/weight or retinal degeneration were observed through 12 months of age. Similarly, our mice did not exhibit any apparent symptoms of splenomegaly, anhidrosis¹⁹, or vision deficits and it remains to be determined if these deficits can occur in our ROSAH mouse model at an advanced age. Unlike the mouse model reported previously by Kozycki et al.³, we were largely unable to detect elevated levels of these same chemokines in systemic circulation with the exception of CXCL9, the upregulation of which was evident in the serum of ROSAH model mice and suppressed by DF-003. This difference may be related to the fact that our mice were heterozygotes as compared to the homozygous ROSAH model mice developed previously³. While our mice thus more closely mimic the heterozygous presentation of human ROSAH syndrome patients, the more muted effects of a single copy of human ALPK1[T237M] may have translated to serum chemokine production at levels below the limits of detection for our ELISAs.”

3)The authors are to be commended for their histologic studies of the retinas of knock-in mice. Are there any comparable data from patients with ROSAH syndrome, especially during the time interval when they are actively losing their vision? Are there any data in ROSAH patients comparable to the gene expression data in the eyes, optic nerves, and brains of the knock-in mice?

This is an important point, and we thank the reviewer for raising it. While similar patient datasets would be invaluable comparators for our study, to our knowledge they are presently unavailable owing to the difficulty

associated with their collection. In our revised Discussion, we now include further discussion of the limitations in our study, as follows:

“A limitation of our study is that ROSAH mice do not show signs of visual acuity loss, and thus it was not possible to determine if the DF-003-mediated alleviation of ocular inflammation in mice could lead to vision improvement. To our knowledge, neither gene expression nor retinal histology has been investigated in ROSAH patients and thus, we cannot conclude whether the observations made in our model mice also apply to humans. Additionally, it remains to be determined whether DF-003 can reduce ocular inflammation in humans as it does in our model, and if so, whether reduced eye inflammation can ultimately lead to the cessation of retinal degeneration in ROSAH patients.”

4) Given the fact that ALPK1 is a relatively newly recognized kinase, the authors may wish to comment on the fact that DF-003 could be used to interrogate the biology of ALPK1 either in mice or in human subjects who receive the agent as part of a clinical trial;

This is an excellent suggestion, and we have added to our Discussion a sentence describing the potential value of DF-003 for interrogating ALPK1 biology:

“With the discovery of the potent, selective ALPK1 inhibitor DF-003, there is also the potential to further interrogate ALPK1 biology through *ex vivo* analyses of ROSAH patient-derived cells, mouse model experiments, and human clinical trials.”

5) The authors may wish to comment on possible reciprocal adoptive transfer experiments in the T237M and wild-type knock-in mice to determine the role of bone marrow-derived cells in the pathogenesis of ROSAH syndrome;

We appreciate this suggestion, as such experiments would be a valuable approach to clarifying cell-type-specific contributions to ROSAH biology. We have revised our Discussion section as follows to mention these experiments as an avenue for future study:

“Based on our results, there are several important avenues for future research. Single-cell transcriptomic and proteomic analyses may be used to identify the cell types responsible for the autoinflammatory status of our model mice. Reciprocal adoptive transfer experiments may also be implemented to study the relative contributions of bone marrow-derived immune cells and tissue resident immune cells in the pathogenesis of ROSAH syndrome.”

6) Both in the Introduction and in the Discussion, the authors allude to the inconsistent efficacy of cytokine inhibitors in preventing vision loss in ROSAH syndrome. While, as a class, it is true that cytokine inhibitors show

inconsistent effects, there are promising early data that tocilizumab may have a beneficial effect (as noted by the authors). However, at this point there are just not enough data at the level of individual cytokine inhibitors to make any definitive positive or negative pronouncement. These points should be clarified in the manuscript;

We fully agree that no specific conclusions can yet be drawn regarding the efficacy of cytokine inhibitors and thank the reviewer for requesting our further clarification on this point. We have revised the Discussion as follows to provide greater context when discussing cytokine inhibitor outcomes in ROSAH patients:

“At present, there is no approved drug for ROSAH syndrome and patients have been typically treated with anti-TNF and IL-1 inhibitors^{3,14-16}. These drugs have been linked to improvements in subjective symptoms of ROSAH, while tocilizumab (anti-IL-6) showed early promising data with reduced ocular inflammation in patients unresponsive to TNF and IL-1 inhibition³. However, formal clinical trials of drugs to treat ROSAH have not yet been initiated, and with limited data, it remains uncertain whether the positive effects of individual anti-cytokine therapy are attributable to drug efficacy or inherent disease variation. Because the role of individual cytokine inhibitors remains untested, we sought to develop a drug that targets the root cause of ROSAH and has the potential to inhibit the production of multiple cytokines caused by disease-causing ALPK1 mutations, with the ultimate goal of bringing this drug forward for testing in a clinical trial.”

7) The authors should consider citing the recent Science paper by Yue Tang and colleagues regarding the coevolution of β -D-manno-heptoses and ALPK1s, since it provides a broader context for the current manuscript;

We appreciate this suggestion. We have incorporated the findings of Yue Tang in our Introduction to provide further context regarding the biology of ALPK1, as follows:

“Recently, ALPK1 was also shown to be activated by similar pathogen-associated molecular patterns (PAMPs), including cytidine diphosphate (CDP)-heptose and uridine diphosphate (UDP)-heptose, both capable of initiating downstream NF- κ B-mediated inflammatory signaling⁸.”

8) On page 5 of the manuscript, I believe that on lines 16 and 19 ‘ALPK1-KD’ should be ‘mAlpk1-KD’.

Thank you for catching this inconsistency - in the revised manuscript, we have changed the appropriate terminology to read ‘mAlpk1-KD’ in all instances.

Reviewer 2

This manuscript by Fan et al reports a small molecule ALPK1 inhibitor as a potential drug candidate for ROSAH syndrome. The absence of approved drugs for the treatment of ROSAH renders this study a useful guide for future research efforts in this direction. The authors designed a nanomolar inhibitor (DF-003) with an adequate pharmacokinetic profile for oral dosing. In addition, the research team performed an extensive panel of screening (in vitro and in vivo) to demonstrate the therapeutic potential of the new compound. However, there are several drawbacks to this work as detailed below that would justify not moving forward with this manuscript for publication in Nature Communications:

1) The design strategy of DF-003 and the details provided on the 2 pharmacophores being combined are insufficient.

In our revised manuscript, we have provided new details regarding the design of DF-003. To address the reviewer's concern, we have revised our manuscript by including new text regarding the production of Pharmacophores 1 and 2 in Supplementary Note 1:

“One of these compounds, termed Hit 1, with an IC_{50} of 4 μ M in a TR-FRET (time-resolved Förster resonance energy transfer)-based ALPK1 kinase assay, emerged as a promising starting point for further investigation (Supplementary Note Fig. 1D).

Using Hit 1 as a starting point, a series of urea analogs (Supplementary Note Tables 2-4) was synthesized and evaluated. In general, the urea analogs of Hit 1 did not significantly impact potency (Supplementary Note Table 2). However, the results shown in Supplementary Note Tables 3-4 suggested that significant molecular space was available for SAR development in the region of attachment of the benzylpiperazine moiety or alkyl alcohol moiety. Further SAR studies of the five-member ring between the phenyl and thiazole group of Hit 1 showed that when this moiety was replaced with smaller rings, such as a three-member ring (**7**) or four-member ring (**8**), a decrease in potency was observed. However, the five-member ring could tolerate being opened, and substitution of lipophilic groups was preferred. The SAR study on the opened five-member ring showed that when the R_2 group on the methylene position between the phenyl and thiazole group (R_2/R_3) was larger, this had a detrimental effect on potency (**70**). A decrease in potency was also noted with the R_2 substitution of H (**59**). This phenomenon indicated that the presence of a suitably sized lipophilic group at this methylene position is crucial for potency. Lipophilic substitutions such as methyl-ethyl or methyl-methyl combinations were associated with improved ALPK1 inhibitory activity. Other alkyl group combinations were also investigated, including ethyl-ethynyl, methyl-propynyl, methyl/methoxypropynyl, and methyl-ethynyl, the latter of which was associated with the best potency (Supplementary Note Tables 4, 7). Substitution on the phenyl ring of Hit 1 (Ring A) was additionally investigated. While the substitution of CN or OCF_3 groups resulted in

decreased potency (Supplementary Note Table 3), halogen, CH₃, and OMe groups exhibited approximately equal activity levels (Supplementary Note Table 3). This SAR study indicated that the presence of an electron donating group was associated with more favorable activity than an electron withdrawing group. Compounds **5** and **6** clearly showed that *meta*-substitution negatively impacted inhibitor potency. SAR development around Hit 1 yielded the optimized compound **34** (IC₅₀ = 48 nM) containing Pharmacophore 1 (Supplementary Note Table 4 and Supplementary Note Fig. 1D).

In a second high-throughput screen, a separate small molecule library of ~200,000 compounds (from WuXi AppTec) was screened using the TR-FRET ALPK1 kinase assay. Compound **23** (Supplementary Note Table 3) at 20 μM was used as a positive control in the screen. From this screen, 14 hits were identified. The IC₅₀ values of these hits in the TR-FRET kinase were below 20 μM, and all hits exhibited drug-like structures. Hit 2 (IC₅₀ = 11 μM) was selected as being desirable for follow-up studies (Supplementary Note Fig. 1D). Hit 2 was further modified in an extensive SAR study (Supplementary Note Table 5) which revealed that the methoxy group at benzyl-thiazole position 7 adversely impacted inhibitory activity of the resultant compound. When this 7-methoxy substitution was removed (**35**), potency levels increased significantly (Supplementary Note Table 5). Subsequent SAR work focused on the para substitution (R₇) of the di-F phenyl ring of Hit 2. The results demonstrated that a basic group is crucial to improve potency. SAR optimization of Hit 2 led to the determination that piperazine substitution at the R₇ position of the di-F phenyl ring yielded the compound with the best activity (**43**), IC₅₀ = 209 nM, containing Pharmacophore 2 (Supplementary Note Table 5 and Supplementary Note Fig. 1D). We conducted a further SAR study of Pharmacophore 1 by synthesizing a series of amide compounds (Supplementary Note Table 6). Finally, the combination of Pharmacophore 1 and Pharmacophore 2, as shown in Supplementary Note Fig. 1D and Supplementary Table 6, yielded DF-003 (**50**), which provided the most potent inhibition of ALPK1. In order to conduct more in-depth SAR analyses and to further optimize DF-003, another array of compounds was synthesized around the Pharmacophore 1 portion of DF-003 (Supplementary Note Table 7). DF-003 was ultimately selected as a compound for further drug development based on its good inhibitory activity and other desirable development properties including its pharmacokinetics, drug metabolism, and toxicology profile.”

2) There is no co-crystal structure reported to elucidate the binding mode of DF-003.

We thank the Reviewer for bringing up this important point and have addressed the issue in the following text from new Supplementary Note 1 as follows:

“To aid in the interpretation of SAR results we built a structure-based modeling application to understand the interactions of ALPK1 inhibitors with the binding pocket in this enzyme. A single crystal X-ray structure of the ALPK1 N-terminal domain (ALPK1-ND), ALPK1 kinase domain (mouse Alpk1-KD; mAlpk1-KD), and the agonist ligand ADPH was successfully generated. To date, the human ALPK1-KD has not been amenable to crystal growth. While we attempted to co-crystalize ALPK1 inhibitors with the wild-type ALPK1 kinase domain, we

were not successful.

Nonetheless, as mouse and human ALPK1-KD are highly conserved, sharing 89% amino acid sequence identity, we did successfully co-crystallize the human ALPK1-ND (residues 1-446) with the mAlpk1-KD (residues 949-1231) in the presence of ADPH. The 2.25 Å structure of the complex was determined using selenomethionine-based single-wavelength anomalous dispersion (SAD) (Methods, *Crystallography*). The ALPK1-ND consists of 18 alpha helices (Supplementary Note Fig. 2A, B) and interacts with mAlpk1-KD in the co-crystal structure. Alpha-helices 14, 16, and 17 of the ALPK1-ND interact with the N-lobe of the mAlpk1-KD. The linking regions for α -helices 7, 8, 9, and 10 of the ALPK1-ND were found to interact with the C-lobe of mAlpk1-KD. ADPH was observed in a pocket of ALPK1-ND (Supplementary Note Fig. 2A, B). As described in the Methods, we developed a human homology KD model and conducted docking experiments to understand the interaction between DF-003 and the ALPK1 kinase domain. Much like the binding pocket established for an alpha-kinase reference protein, 3PDT, the structure of which was determined by single crystal X-ray structure analysis¹, the three consecutive lysine (LYS) residues (LYS 1041, LYS 1042, and LYS 1043) in the p-loop structure of ALPK1 help form the unique molecular binding pocket of ALPK1 (Supplementary Note Fig. 3A).”

3) The docking studies performed to study the potential binding mode of DF-003 are minimal.

We agree that further clarification of the binding mode of DF-003 is an important addition to the manuscript. In addition to the molecular docking of DF-003 with ALPK1, we also performed molecular dynamics simulations of the DF-003-ALPK1 complex to verify the stability of this complex, as well as molecular docking of compounds similar to DF-003 with ALPK1 to verify the correspondence between the model and the compound SAR. We have expanded the revised manuscript with new data pertaining to this point. The relevant text is below:

“As described in the Methods, we developed a homology model of mAlpk1-KD and conducted docking experiments to understand the interaction between DF-003 and the ALPK1 kinase domain. Much like the binding pocket established for an alpha-kinase reference protein, 3PDT, the structure of which was determined by single crystal X-ray structure analysis¹, the three consecutive lysine (LYS) residues (LYS 1041, LYS 1042, and LYS 1043) in the p-loop structure of ALPK1 help form the unique molecular binding pocket of ALPK1 (Supplementary Note Fig. 3A). This unique attribute may explain the high kinase selectivity of our inhibitor DF-003. DF-003 was docked with mAlpk1-KD, and an analysis of the docking results revealed that DF-003 exhibited a high docking score (-8.493 kcal/mol), forming stable interactions with the key residues of the active site of ALPK1 (Supplementary Note Fig. 2C, D). The interactions formed by DF-003 included two hydrogen bonds (with GLY 1136 and GLU 1137), two π - π stacked interactions (with TYR 1133 and PHE 1138), and one salt bridge interaction with GLU 1137. The hydrogen bond between amide NH and GLY 1136 and π - π interactions ensure a

hinge binding mode similar to that of ATP (Supplementary Note Fig. 3B and C), which contributes to the stability of the binding, while the piperazine salt bridge and hydrogen bond interactions further enhance its binding affinity and molecular interactions (Supplementary Note Fig. 2D). Even though the interaction of the di-fluorine (di-F) substitution with an amino acid in the kinase domain pocket was not observed, as reported, the size of fluorine-containing functional groups is very unique². The size of fluorine groups can help direct a compound into its target pocket. Once the molecule is inside the binding pocket, the electronic characteristics of the fluorine group can then influence other binding moieties on the molecule to enhance the binding potency of the drug². In our case, the di-F substitution might force the phenyl ring into a more suitable binding pose. When the compound lacks the di-F moiety, an obvious reduction in potency was noted (45). In addition, the electronic effect of the di-F group may affect the electron distribution of the piperazine ring, providing a better interaction with GLU 1137 in the kinase pocket (Supplementary Note Fig. 4A). Although the interaction of the methyl-ethynyl group with the protein pocket was not observed in this molecular modeling effort, based on the results of the SAR study we concluded that ethynyl was a preferred lipophilic group conducive to a good compound fit in the deep kinase domain pocket (Supplementary Note Fig. 4B). Docking scores and kinase inhibition activity for the SAR study compounds that led to the development of DF-003 were strongly correlated (Supplementary Note Tables 2-7 and Supplementary Note Fig. 4C). The molecular dynamics of the ALPK1-DF-003 complex also revealed good stability, with a protein root mean square deviation (RMSD) < 3 Å (Supplementary Note Fig. 4D).”

4) Site-directed mutagenesis was not performed to verify the key binding residues predicted to interact with DF-003

To date, we have not used site-directed mutagenesis to produce ALPK1-KD isoforms for the purposes of identifying key binding residues. Thus, the binding residues currently mentioned in the manuscript are predicted based on docking results from the binding pocket model. While we agree that the suggested mutagenesis experiments will be an important aspect of the full characterization of the ALPK1-DF-003 interaction in the future, we believe that at this time they are not essential to support the preclinical efficacy of DF-003 reported in the current version of the study.

5) Since the authors used thermal shift assay to identify hits, I expected extensive MST, SPR, and ITC data to validate the binding of the compound to the target.

We agree and have included additional new details describing the validation of DF-003 binding to ALPK1 and have revised the manuscript accordingly. We used an SPR approach to validate hits (AS-252424, sulfasalazine, TBB, eltrombopag, novobiocin, niflumic acid) identified in a pilot screen using the thermal shift assay. Of these 6 hits, all were validated in SPR

experiments performed with a Biacore 3000 instrument, and SPR data for AS-252424 (as an example) has been incorporated in the revised manuscript (Supplementary Note Fig. 1A).

Hits from our thermal shift assay pilot screen and high-throughput screen were tested using an *in vitro* ALPK1 kinase assay, after which p-TIFA levels were measured by Western blotting. 18 out of 243 TSA hits exhibited an inhibitory or activating effect on ALPK1 kinase activity, suggesting a real interaction with the ALPK1 protein. The Hit 1 kinase assay results (as an example) are now shown in the new Supplementary Note, Figure 1B.

We believe these results support the reliability of the thermal shift assay as an approach to identifying ALPK1 binders. The relevant text is below:

“In a pilot TSA screen using a generic commercial kinase inhibitor library and an FDA-approved drug library (total of 1680 compounds; EFEBIO), AS-252424 (Supplementary Note 1 Fig. 1A) was discovered to cause an increase in melting temperature (T_m) of 3.4°C. Surface plasmon resonance (SPR) analyses confirmed the reliability of using a TSA as a means of screening for ALPK1 binders. AS-252424 was found to induce an SPR signal on chips coated with full-length ALPK1 or the ALPK1 kinase domain (Supplementary Note Fig. 1B), suggesting that AS-252424 binds to the kinase domain of ALPK1. SPR was performed on other hits, including sulfasalazine, TBB (tetrabromobenzotriazine), eltrombopag, novobiocin, niflumic acid, and tolcapon to validate the TSA method and the data were summarized in Supplementary Note Table 1.”

6) No sufficient evidence for cellular target engagement of DF-003 is provided.

We appreciate the suggestion to provide more information regarding evidence for the cellular target engagement of DF-003, and have incorporated new data into the revised manuscript. In response to the recognition of ADPH, DF-006, or other agonists, ALPK1 phosphorylates TIFA and triggers subsequent TIFAsome assembly and NF- κ B activation. Our data show that DF-003 can potentially inhibit ALPK1 agonist DF-006-induced pro-inflammatory gene expression. To better demonstrate target engagement, we examined the assembly of TIFAsomes, which represent TIFA phosphorylation status in cells, upon DF-006 activation with or without DF-003 treatment. DF-003 suppressed the formation of TIFAsomes induced by DF-006 in HEK293 cells, suggesting that intracellularly it directly exerts its anti-inflammatory function through the inhibition of ALPK1 kinase activity. This new data has now been incorporated into the revised manuscript (Figure 2B, C). The associated text is below:

“Once phosphorylated, TIFA dimers engage in intermolecular interactions and trigger the assembly of large TIFAsomes⁹. DF-003 potentially suppressed DF-006-induced TIFAsome

formation (Fig. 2b, c), suggesting its intracellular inhibition of ALPK1.”

7) Details regarding the thermal shift assay and TR-FRET (assay development, miniaturization, controls used) are minimal and insufficient to reproduce the work.

We appreciate the suggestion to provide more detail to ensure that anyone can reproduce our experiments and have modified the manuscript accordingly. New details of the thermal shift and TR-FRET assays, including the composition of the reaction mixture as well as all proteins and reagents used have been added to the revised Methods section under the subtitles *‘High-throughput screening using a thermal shift binding assay’*, *‘TR-FRET kinase assay’*, and *‘High-throughput screen with TR-FRET kinase assay’*.

During assay development, the thermal shift assay was validated using ATP and ADP as positive controls, which yielded ALPK1 T_m increases of 1.1° C and 1.4° C, respectively. On each 384-well screening plate, 16 repeats of negative control (compound solvent, protein, and dye alone, providing the T_m of ALPK1) and 16 repeats of positive controls (AS-252424) were included. Hits from TSA screening were subsequently validated in a Western blot-based kinase reaction.

Kinase reaction conditions were optimized in a TR-FRET assay such that there was a positive correlation between TR-FRET signal and enzyme quantity or reaction time in the absence of compounds. In high-throughput TR-FRET screening, on each screening plate, multiple repeats of AS-252424 (final concentration: 20 μ M) were included in the kinase reaction as a positive control.

8) A med chem campaign focused on the optimization of DF-003 was not pursued.

In our revision of the manuscript, we have reported further SAR around the development candidate DF-003 (Supplementary Note Table 7). For further details, please refer to our response to Reviewer 2 - Question 1.

Reviewer 3

This is an interesting study that has generated new inhibitors of the ALPK1 protein. My review focusses on the protein structure and docking aspects of the study.

The crystal structure of ALPK1/ADPH is of good quality overall, with geometry

and refinement statistics consistent with a 2.2Å resolution. Crucially, the binding pose of the ligand is well-evidenced by clear electron density, and it has good geometry. The kinase part of the structure, and the interface between the two domains are novel findings. The authors could increase the impact of their paper for molecular/structural biologists in the kinase field with a small amount of extra effort to include a supplementary figure showing:

i) A comparison of the kinase domain structure with the most similar known kinase (i.e. PDB 3PDT)

We appreciate this valuable suggestion and have revised the text accordingly. Specifically, we performed an alignment of the ALPK1 binding pocket and the 3PDT binding pocket. Relative to the reference protein (3PDT), the three consecutive lysine residues (LYS 1041, LYS 1042, and LYS 1043) in the p-loop structure of ALPK1 were found to constitute the molecular binding pocket of ALPK1 (see new Supplementary Note Fig. 3A). We have revised the main text and Supplementary information for our article to reflect these changes.

ii) Details of the molecular interactions at the interface between the N-terminal domain and kinase domain.

We thank the reviewer for this suggestion. We have added new details of the molecular interaction at the interface between the N-terminal domain and kinase domain to Supplementary Note 1 in our revised manuscript as follows:

“The ALPK1-ND consists of 18 alpha helices (Supplementary Note Fig. 2A, B) and interacts with mAlpk1-KD in the co-crystal structure. Alpha-helices 14, 16, and 17 of the ALPK1-ND interact with the N-lobe of the mAlpk1-KD. The linking regions for α -helices 7, 8, 9, and 10 of the ALPK1-ND were found to interact with the C-lobe of mAlpk1-KD. ADPH was observed in a pocket of ALPK1-ND (Supplementary Note Fig. 2A, B).”

The binding mode of DF-003 and its precursors was investigated using molecular docking. The binding site overlaps with that of ATP, consistent with the biochemical data. However, there are a few issues with the modelling that must be addressed:

i) More information on the homology modelling and molecular docking should be provided (what software was used, what were the key parameters/settings used in the software, what metrics were used to evaluate the reliability of the homology model, are there any alternative binding modes of DF-003 that generated a similar docking score, say within 1 kcal/mole?)

Thank you for this important suggestion that has now been incorporated into the revised manuscript in the Methods section entitled “Homology modeling

and molecular docking” . For these analyses, we used the Schrödinger Maestro Suite 2021-02 software. Homology modeling was performed with the Prime module following standard knowledge-based model-building methods under the OPLS4 force field. For molecular docking analyses, Maestro and the Glide module were used with the following docking parameters: FORCEFIELD OPLS_2005, Standard precision mode, setting the hydrogen bond interaction of hinge binding as a constraint.

Molecular dynamics simulations of the DF003-ALPK1 complex were used to verify the stability of this complex, and the molecular docking of compounds similar to DF-003 with ALPK1 was used to verify the correspondence between the model and the compound SAR. The RMSD of protein and ligand in molecular dynamics simulations, as well as the correlation between docking scores and compound activity, are two metrics that were used to evaluate the reliability of the homology model. Multiple binding poses were docked into the model to assess the effect of minor fit-to-site changes on key molecular interactions. Similar docking scores from these poses suggested the expected interactions were occurring.

ii) The docking score is comparable with that of ATP, but how does the binding affinity of ATP compare with that of DF-003? If there is a large difference in their affinities, then the similarity of the docking score suggests the method is not fully reliable.

According to Schrödinger Knowledge Base (<https://www.schrodinger.com/kb/572>) and GOLD Online Support (<https://www.ccdc.cam.ac.uk/support-and-resources/support/case/?caseid=83b3a474-5043-43c2-9c0c-fc481648bce4>), docking scores are an estimate of the binding energy, and are optimized for the prediction of ligand binding positions rather than the prediction of binding affinities. The particular receptor conformation used for docking might not be optimal for all binding actives. Docking can be used to predict binding affinity for new members in the same series.

For the same series of compounds, the receptor binding conformation is highly similar, so the docking score can be used to approximate the affinity of the compound. However, for different types of compounds, the binding conformation of receptors is different, and interaction is very different. The level of affinity cannot be directly represented by the docking score. In light of these considerations, we believe that it is inappropriate to compare the docking scores of DF-003 and ATP in the article, so we have removed the

relevant text (highlighted in red and struck through in the following sentence) from our revised manuscript:

“Analysis of the docking results revealed that DF-003 exhibited a high docking score (-8.493 kcal/mol) ~~consistent with that of ATP (-8.447 kcal/mol)~~, forming stable interactions with the key residues of the active site in ALPK1”

iii) The overall binding mode is reasonable, but some details of the molecular recognition are surprising. Specifically, there are no contacts between the protein and the difluorophenyl group or the ethynyl group. This may reflect small errors in the docking (which may or may not have accommodated induced fit) or possibly an incorrect binding pose. The authors can address this point by explaining how the model is supported by compound SAR (mentioned in the text but no data shown) and by carrying out further docking/molecular dynamics with DF-003 to investigate whether additional, potentially transient interactions are made.

We thank the Reviewer for raising this point and we have addressed it in our new Supplementary Note 1, demonstrating the potency and selectivity of DF-003 as an ALPK1 inhibitor as follows:

“Even though the interaction of the di-fluorine (di-F) substitution with an amino acid in the kinase domain pocket was not observed, as reported, the size of fluorine-containing functional groups is very unique². The size of fluorine groups can help direct a compound into its target pocket. Once the molecule is inside the binding pocket, the electronic characteristics of the fluorine group can then influence other binding moieties on the molecule to enhance the binding potency of the drug². In our case, the di-F substitution might force the phenyl ring into a more suitable binding pose. When the compound lacks the di-F moiety, an obvious reduction in potency was noted (45). In addition, the electronic effect of the di-F group may affect the electron distribution of the piperazine ring, providing a better interaction with GLU 1137 in the kinase pocket (Supplementary Note Fig. 4A). Although the interaction of the methyl-ethynyl group with the protein pocket was not observed in this molecular modeling effort, based on the results of the SAR study we concluded that ethynyl was a preferred lipophilic group conducive to a good compound fit in the deep kinase domain pocket (Supplementary Note Fig. 4B). Docking scores and kinase inhibition activity for the SAR study compounds that led to the development of DF-003 were strongly correlated (Supplementary Note Tables 2-7 and Supplementary Note Fig. 4C). The molecular dynamics of the ALPK1-DF-003 complex also revealed good stability, with a protein root mean square deviation (RMSD) < 3 Å (Supplementary Note Fig. 4D).”

In support of the above discussion around the fluorine and ethynyl substituents, in the new Supplementary Note portion of the revised manuscript, we have added substantial detail regarding how the binding-site model is supported by compound SAR (see Supplementary Note Tables 1-7).

Reviewer Responses

Reviewer #1:

I am satisfied with the authors' response to my comments, with one small exception. I think that in response to my comment #7, the authors should insert 'across multiple kingdoms' after '(PAMPs)', thus:

Recently, ALPK1 was also shown to be activated by similar pathogen-associated molecular patterns (PAMPs) across multiple kingdoms, including .

We have updated the text of the Introduction to incorporate the suggested phrase "across multiple kingdoms", which is highlighted in red in our revised manuscript.

Reviewer #2:

- 1. SPR analysis was not performed for DF-003**
- 2. SPR sensograms need to show the impact of the multiple doses used to calculate the K_d rather than a single-dose.**

We thank the reviewer for the suggestion of using SPR analysis to demonstrate DF-003's binding to ALPK1 kinase domain. We have performed SPR analysis and the multi-dose sensogram has been added to Supplementary Note 1, Figure 1d. To measure DF-003's binding affinity to ALPK1 in its kinase active mode, we used full-length ALPK1 with ADP-heptose added to the system. The calculated K_D is 90 nM.

- 3. The authors make a point that site-directed mutagenesis are not essential to validate the preclinical efficacy for DF-003. However, these studies will validate the computational predictions by the authors, which is a major part of the manuscript.**

We have made diligent efforts to execute site-directed mutagenesis analyses to validate computational predictions. The interaction between DF-003 and ALPK1 is predicted as follows:

"The interactions formed by DF-003 included two hydrogen bonds (with GLY1136 and GLU1137), two π - π stacked interactions (with TYR1133 and PHE1138), and one salt bridge interaction with GLU1137 (Supplementary Note Fig. 3b)."

In the current revised manuscript, the results of site-directed mutagenesis analyses have been added to Supplementary Note 1 as follows:

"To validate the molecular homology modeling, we tested whether the inhibitory potency of DF-003 would be altered using different mutant ALPK1 kinases. Site-directed mutagenesis was used to create single amino acid mutants of ALPK1 at positions predicted to mediate binding

with DF-003 (Supplementary Note 1, Table 8). First, we tested whether expressing these mutant proteins in ALPK1 knock-out HEK293 cells could rescue ALPK1 agonist-induced NF- κ B activation. Co-transfection of an NF- κ B reporter construct and measurement of luciferase activity after addition of ALPK1 agonist DF-006 was used to assess NF- κ B activation. GLU1137A, GLU1137K, and TYR1133A mutants retained at least partial ALPK1 activity, and we purified those mutant enzymes and measured the IC₅₀ of DF-003 compared to wild-type ALPK1. DF-003 exhibited increased IC₅₀ values of 44.6 nM, 23.7 nM, and 29.0 nM for GLU1137A, GLU1137K, and TYR1133A, respectively, in contrast to an IC₅₀ of 9.2 nM for the wild-type enzyme, suggesting the reduced binding affinity of DF-003 to the three mutants (Supplementary Note 1, Table 8). These results are in line with the prediction from the homology modeling that GLU1137 and TYR1133 are involved in the binding of DF-003 to ALPK1.

Single amino acid substitution mutants at GLY1136 and PHE1138 failed to rescue the loss of NF- κ B activation by DF-006 in ALPK1 knock-out cells, suggesting these mutants were kinase dead, likely because these residues are also critical for binding of ATP (Supplementary Note 1, Fig. 3). The result was not surprising as DF-003 is an ATP-competitive inhibitor and binds to the catalytic site of ALPK1. The residues predicted to interact with DF-003 in our computational model (TYR1133, GLY1136, GLU1137, and PHE1138) are either very close to catalytic amino acid residues or are catalytic residues themselves. Similarly, alpha kinase MHCK-A lost function when the conserved catalytic site PHE was mutated¹. The single amino acid mutant of GLY1136 also led to loss of kinase activity, likely because it is also involved in hydrogen bonding between ALPK1 and ATP (Supplementary Note Fig. 3d). GLY1136 is also a conserved residue across all identified alpha kinases². “

The experiment results were also summarized in Supplementary Note 1, Table 8:

	Change to	NF- κ B activation after the addition of DF-006 when expressed in ALPK1 knock-out cells (% of cells expressing WT ALPK1)	IC ₅₀ of DF-003 in kinase assay (nM)
WT		100%	9.2
TYR1133	A	19%	29.0
	Q	0	Kinase not purified
PHE1138	A	0	Kinase not purified
	Q	0	Kinase not purified
GLU1137	A	36%	44.6
	K	49%	23.7
GLY1136	A	0	Kinase not purified

References

1. Yang Y, Ye Q, Jia Z, Cote GP. Characterization of the Catalytic and Nucleotide Binding Properties of the alpha-Kinase Domain of Dictyostelium Myosin-II Heavy Chain Kinase A. *J Biol Chem*. 2015;290:23935-23946. doi: 10.1074/jbc.M115.672410
2. Middelbeek J, Clark K, Venselaar H, Huynen MA, van Leeuwen FN. The alpha-kinase family: an exceptional branch on the protein kinase tree. *Cell Mol Life Sci*. 2010;67:875-890. doi: 10.1007/s00018-009-0215-z